# Defect-induced triple synergistic modulation in copper for superior electrochemical ammonia production across broad nitrate concentrations

Bocheng Zhang[1,2], Zechuan Dai[1,2], Yanxu Chen[1,2], Mingyu Cheng[1], Huaikun Zhang[1], Pingyi Feng[1], Buqi Ke[1], Yangyang Zhang[1] & Genqiang Zhang [1] ✉

Nitrate can be electrochemically degraded to produce ammonia while treating sewage while it remains grand challenge to simultaneously realize high Faradaic efficiency and production rate over wide-range concentrations in real wastewater. Herein, we report the defect-rich Cu nanowire array electrode generated by in-situ electrochemical reduction, exhibiting superior performance in the electrochemical nitrate reduction reaction benefitting from the triple synergistic modulation. Notably, the defect-rich Cu nanowire array electrode delivers current density ranging from 50 to 1100 mA cm$^{-2}$ across wide nitrate concentrations (1–100 mM) with Faradaic efficiency over 90%. Operando Synchrotron radiation Fourier Transform Infrared Spectroscopy and theoretical calculations revealed that the defective Cu sites can simultaneously enhance nitrate adsorption, promote water dissociation and suppress hydrogen evolution. A two-electrode system integrating nitrate reduction reaction in industrial wastewater with glycerol oxidation reaction achieves current density of 550 mA cm$^{-2}$ at −1.4 V with 99.9% ammonia selectivity and 99.9% nitrate conversion with 100 h stability, demonstrating outstanding practicability.

Nitrate is widely found in industrial wastewater and groundwater due to the excessive use of chemical fertilizers and industrial contamination[1–3]. Excessive nitrates can disrupt the global nitrogen balance, and their incomplete conversion to nitrites in the body may cause cancer in humans[4–6]. Electrochemical reduction of nitrate to non-toxic nitrogen is considered an environmentally friendly route with mild reaction conditions[7,8]. However, this method is energy-intensive and lacks value-added benefits[9]. Ammonia, as a multifunctional chemical, is used in the manufacture of fertilizer and medicine, and is a vital carrier of clean energy[10–13]. Electrochemical nitrate reduction (NO$_3$RR) can be employed for distributed production of

green ammonia under mild conditions and driven by locally generated clean energy, which has garnered widespread attention[14–17].

Nitrate concentrations in waste streams span a broad range, from 1 to 100 mM[18]. However, most studies have focused on performance under high nitrate concentrations and theoretical conditions, struggling to achieve satisfactory performance in scenarios with varying nitrate concentrations, particularly under low concentrations where strong HER is present[10,19,20]. The scientific community has increasingly recognized the significance of low-concentration nitrate reduction. For instance, Kang et al. made a significant contribution by developing a Ni$_3$Fe LDH/Cu foam for the reduction of low-concentration nitrate,

[1]Hefei National Research Center for Physical Sciences at the Microscale, CAS Key Laboratory of Materials for Energy Conversion Department of Materials Science and Engineering, University of Science and Technology of China, Hefei, Anhui 230026, China. [2]These authors contributed equally: Bocheng Zhang, Zechuan Dai, Yanxu Chen. ✉e-mail: gqzhangmse@ustc.edu.cn

achieving a commendable 96.8% FE with 65 mA cm$^{-2}$ at 5 mM[20]. Nonetheless, the performance experienced a dip at 1 mM and 2 mM, recording 44% and 61% FE at 10 mA cm$^{-2}$ and 20 mA cm$^{-2}$, respectively, signaling an opportunity for further optimization. Additionally, Junqueira et al. achieved a notable enhancement in FE across a wide concentration range of 1 to 100 mM using a synthesized CuCoSP catalyst, a notable advancement in this field[21]. However, the current density, ranging from 8 to 265 mA cm$^{-2}$, could be a limiting factor for its practical application. Simultaneously, the research on reduction of nitrate to produce ammonia in authentic wastewater is rare and challenging[18]. It is evident that, at low nitrate concentrations, vigorous hydrogen evolution restricts high FE and yield[22]. At typical nitrate concentrations, the insufficient supply of active *H may limit the achievement of industrial-level current[19]. Therefore, we aim to concurrently address the issues of strong hydrogen evolution at low concentrations and the scarcity of active *H supply at typical nitrate concentrations, thus enabling the efficient reduction of real waste streams with a wide range of nitrate concentrations.

Herein, we present defect-rich Cu nanowire array electrode (V-Cu NAE) synthesized through in-situ electrochemical reduction of Cu$_3$N NWs, achieving current densities ranging from 50 to 1100 mA cm$^{-2}$ across nitrate concentrations ranging from 1 to 100 mM and maintaining over 90% FE for degrading nitrate levels in compliance with World Health Organization (WHO) drinking water standards. Operando Synchrotron radiation-FTIR and density functional theory (DFT) calculations reveal that a triple synergistic modulation, including enhanced nitrate adsorption, promoted water dissociation, and suppressed hydrogen evolution, contributes to the exceptional performance across a broad nitrate concentration range. Furthermore, a two-electrode system integrating nitrate reduction reaction (NO$_3$RR) and glycerol oxidation reaction (GOR) in authentic low-concentration industrial wastewater exhibits 550 mA cm$^{-2}$ at −1.4 V, with exceptional ammonia selectivity as high as 99.9%, superior nitrate conversion of 99.9%, and outstanding stability for 100 h, while successfully achieving high-purity product collection in form of both NH$_4$Cl and NH$_3$·H$_2$O.

## Results

Copper hydroxide nanowires (Cu(OH)$_2$ NWs) were prepared by oxidizing copper foam, which were then sintered under ammonia or hydrogen gas to obtain copper nitride nanowires (Cu$_3$N NWs) and Copper nanowires without defects (Cu NWs). Defective copper nanowire array electrode (V-Cu NAE) was obtained by in-situ electrochemical reduction of Cu$_3$N NWs in potassium sulfate solution (Fig. 1a). The powder X-ray diffraction (XRD) analysis (Fig. 1b) confirmed the successful synthesis of Cu$_3$N NWs and their complete conversion to pure Cu phase after in-situ electrochemical reduction[23,24]. Field-emissions scanning electron microscopy (FESEM) and high-resolution transmission electron microscopy (HRTEM) were used to characterize the morphological transition from Cu(OH)$_2$ NWs to Cu$_3$N NWs, and finally to V-Cu NAE (Fig. 1c, d). The morphology of Cu$_3$N NWs (Supplementary Fig. 1) mirrored that of Cu(OH)$_2$ NWs (Supplementary Fig. 2). Notably, upon electrochemical reduction, V-Cu NAE exhibited bending and formed defect-rich black pits, which marked a clear distinction from Cu$_3$N NWs (Fig. 1e). Figure 1c shows an array of defective copper nanowires, where HRTEM (Fig. 1d) reveals dark pits on their surfaces, signifying agglomerations of copper vacancies[25]. To investigate the nature and distribution of these defects more comprehensively, we utilized Cs-corrected TEM techniques. In our analysis showcased in Fig. 1e, we demarcate Cu vacancies using distinct circles, shedding light on their pervasive presence across the sample[26]. Building upon this visualization, we conducted an intensity contour analysis on the atoms situated between the arrows in regions i, ii and iii, as illustrated in Fig. 1f. Our observations underscore that the intensity in the Cu vacancies and agglomeration areas is discernibly lower than

in the surrounding intact lattice regions, bolstering the assertion of a significant presence of Cu vacancies[27].

Energy-Dispersive X-ray Spectroscopy (EDS) (Fig. 1g) indicated that N$^{3-}$ is completely eliminated after electrochemical reduction, resulting in the transformation of Cu$_3$N NWs to V-Cu NAE. Synchrotron radiation X-ray absorption near edge structure (XANES) (Fig. 1h) and X-ray photoelectron spectroscopy (XPS) (Supplementary Figs. 3, 4) failed to detect any significant signal of N in V-Cu NAE[28,29]. In the XANES of V-Cu NAE (Fig. 1i), the peaks at 931.0 and 950.9 eV were attributed to Cu 2$p_{3/2}$ and 2$p_{1/2}$ of metallic Cu phase[30]. The in-situ Raman analysis clearly illustrates (Fig. 1j and Supplementary Fig. 5) the sequential diminishment and ultimate disappearance of the Cu$_3$N Raman peak, implying the formation of metallic Cu[31–33]. Building upon these analyses, XPS results for the V-Cu NAE showed that the Cu 2$p_{3/2}$ and Cu 2$p_{1/2}$ peaks shifted by 0.13 eV and 0.17 eV, respectively, when compared to those of Cu NWs (Supplementary Fig. 6). Such peak shifts might be attributed to the interaction between the metal vacancies and electrons.

The NO$_3$RR performance of the sample was firstly evaluated by using a 0.5 M K$_2$SO$_4$ electrolyte in a typical three-electrode H-type cell (see Supplementary Fig. 7 for the setup), with the working electrode having an area of 1 cm$^2$ (catalysts loading was 3 mg cm$^{-2}$). The resulting ammonia, nitrite and residual nitrate were detected by ultraviolet-visible (UV/Vis) absorbance spectra (Supplementary Fig. 8)[33,34]. As shown in Fig. 2a, the current density of V-Cu NAE increased significantly with the addition of 200 ppm NO$_3^-$ in the electrolyte, and it is clearly that the V-Cu NAE exhibited the most impressive performance compared with Cu foam, Cu nanowires and Cu$_3$N nanowire arrays. The calculated FE and selectivity of ammonia production for the potential ranging from −0.1 to −0.5 V (vs. RHE) (Fig. 2b) showed a volcanic trend, reaching the best performance at −0.3 V (vs. RHE) with FE of 95.1 ± 2.3%, selectivity of 97.2 ± 1.8% and conversion as high as 95.9 ± 0.8% after 1 h catalysis which is superior compared to that of Cu$_3$N NWs (FE 68.0 ± 2.6%, selectivity 83.1 ± 2.5%, conversion 69.2 ± 3.4%) and Cu NWs (FE 68.5 ± 2.1%, selectivity 52.1 ± 1.9%, conversion 51.1 ± 1.6%). Within the potential range of −0.1 to −0.5 V (vs. RHE), both Cu$_3$N NWs and Cu NWs also exhibited lower FE and selectivity compared to V-Cu NAE (Supplementary Fig. 9). The NH$_3$ production rate of the V-Cu NAE was calculated to be 7853 ± 242 μg h$^{-1}$ cm$^{-2}$ at −0.3 V during 1 h of electrolysis, which is much higher than that of the comparing samples (Fig. 2d). Furthermore, the stability of the catalyst can be well evidenced by repeated cycle measurement at −0.3 V for each 1 h cycle, where no visible decay can be observed for FE, selectivity and ammonia production rate within eight cycles (Fig. 2e).

To evaluate the practicability, the nitrate removal ability of the catalyst was conducted at −0.3 V on simulated wastewater containing 200 ppm NO$_3^-$-N. As shown in Fig. 2f, the concentrations of both NO$_3^-$ and NO$_2^-$ could be degraded to 2.1 ppm and 0.52 ppm after 90 mins, which is much lower than WHO drinking water standards (NO$_3^-$-N < 11.3 ppm and NO$_2^-$-N < 0.91 ppm). Importantly, the FE remained above 90% and the nitrate conversion rate can reach 99% with a high selectivity of 99.7% for ammonia production, which demonstrates the excellent nitrate wastewater treatment capacity. Given the fact that the NO$_3^-$ levels in waste streams under different scenario may vary from 1 to 1000 mM, it is necessary for the catalyst to exhibit excellent performance across a wide range of NO$_3^-$ concentrations[18]. Of the various waste streams, textile wastewater and industrial wastewater, with NO$_3^-$-N concentrations ranging from 10 to 50 mM, present a wide range of sources and high utilization value[18]. However, a low NO$_3^-$ concentration might lead to low current density and strong competitive HER. Consequently, we first simulated the degradation of wastewater with NO$_3^-$-N concentrations ranging from 10 to 50 mM. As shown in Fig. 2g, the V-Cu NAE can deliver current density ranging from 200-800 mA·cm$^{-2}$ in the NO$_3^-$-N concentration ranges of 10 to 50 mM at −0.3 V, while maintaining over 90% FE as nitrate was reduced to the

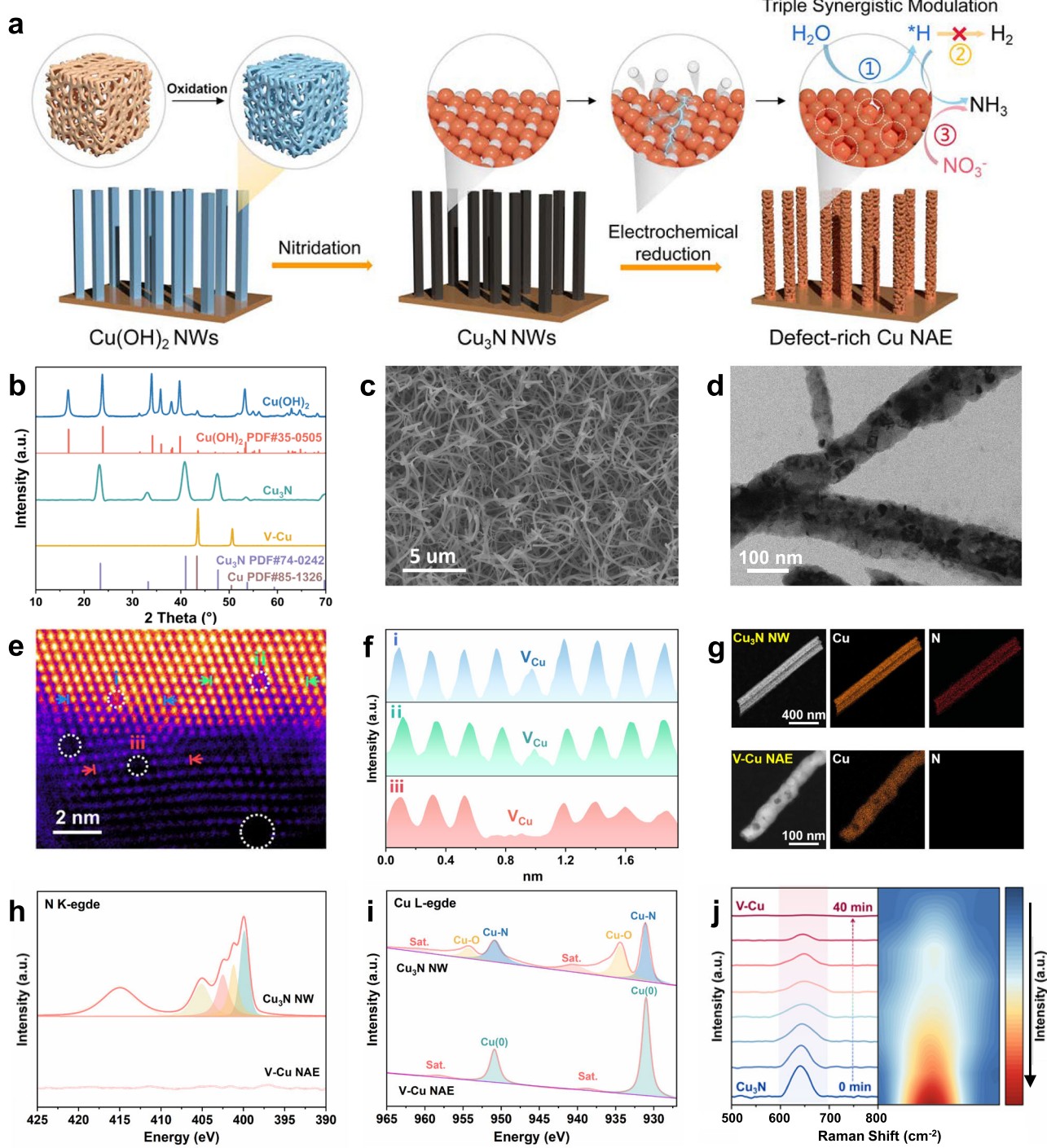

**Fig. 1 | Schematic formation process, morphological and structural characterizations of the V-Cu NAE. a** Schematic illustration of the formation process. **b** XRD patterns, (**c**) SEM, (**d**) HRTEM, (**e**) double Spherical Aberration-corrected HAADF-STEM image of V-Cu NAE, with the circled areas indicating Cu vacancies. **f** Intensity profile of the Cu atoms between the two arrows in regions i, ii and iii of (**e**). **g** EDS mapping of Cu₃N NWs and V-Cu NAE. Synchrotron radiation soft XANES of Cu₃N NWs and V-Cu NAE: (**h**) N K-edge spectra and (**i**) Cu L-edge spectra. **j** In situ Raman spectroscopy of the electrochemical reduction of Cu₃N to V-Cu and its contour map corresponding to the 600–700 cm⁻¹ range. Color scale with arrow indicating the change in signal intensity from low to high.

concentration range of drinking water. Furthermore, even at ultralow nitrate concentration (polluted ground water) at −0.2 V, the V-Cu NAE exhibited high nitrate conversion and FE. Meanwhile, under high nitrate concentrations (0.1 to 1 M liquid nuclear waste) at −0.4 V, the V-Cu NAE displayed high current density of 1000-1300 mA·cm⁻² and more than 95% FE as nitrate was reduced to the concentration range of drinking water level (99.8% NO₃⁻ conversion). The ampere-level current could be attributed to defects promoting H₂O dissociation to

form active *H, which is essential for hydrogenation to produce ammonia. In conclusion, the catalyst effectively removed nitrate from various nitrate sources with FE exceeding 90% at high current density, signifying considerable potential for practical application. In comparison, the Cu NWs was suboptimal (Supplementary Fig. 10). Over wide-range concentrations, all metrics − current density, FE, selectivity, and conversion rate − for Cu NWs lagged behind those of V-Cu NAE. This distinction underscores defects as a key determinant in optimizing

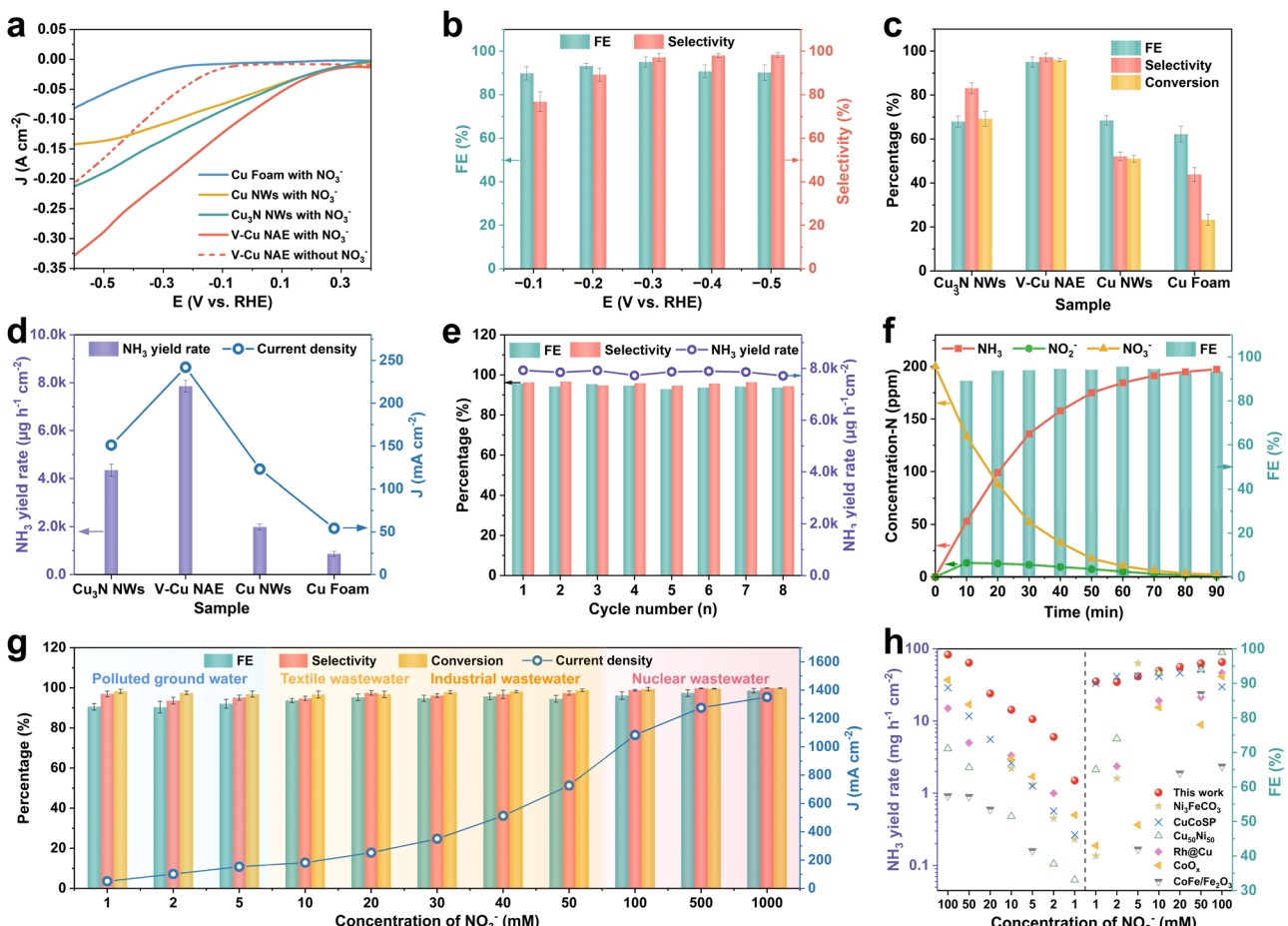

**Fig. 2 | The electrocatalytic activity for NO₃RR in H-cell. a** *j-E* curves of Cu foam, Cu NWs, Cu₃N NWs and V-Cu NAE in 0.5 M K₂SO₄ electrolyte (pH=6.92 ± 0.01) with (solid lines) and without (dotted line) 200 ppm NO₃⁻-N. The scan rate for LSV was 5 mV s⁻¹. All reactions were conducted at a temperature of 25 °C with a magnetic stirring speed of 300 rpm. The measured resistance of the electrochemical cell was 1.63 ± 0.03 Ω. **b** Potential-dependent Faradaic efficiency (FE, green bars) and selectivity of ammonia (green bars) over V-Cu NAE (catalysts loading was 3 mg cm⁻²). **c** FE and selectivity of ammonia and conversion rate of nitrate (yellow bars) over different samples. **d** The ammonia yield rate (purple bars) of as-prepared samples at −0.3 V (vs. RHE). **e** The cyclic stability test of V-Cu NAE. (f) Time-dependent concentration change of NO₃⁻, NH₃, NO₂⁻ and FE over the V-Cu NAE at −0.3 V (vs. RHE). **g** Current density, FE of NH₃, selectivity of NH₃, and conversion of NO₃⁻ in different concentration nitrate sources (contaminated groundwater, textile wastewater, industrial wastewater, liquid nuclear wastes). **h** Comparison of ammonia yield and Faraday efficiency of V-Cu NAE with other catalysts at different nitrate concentrations. Error bars represent the standard deviations calculated from three independent measurements.

performance over wide-range concentrations. As shown in Fig. 2h, within the nitrate concentration range of 1–100 mM, the V-Cu NAE outperformed other catalysts by yielding a notably higher ammonia production rate while preserving an FE equal to or exceeding that of competing catalysts. This demonstrated the exceptional performance of the V-Cu NAE across a wide range of nitrate concentrations, providing a potential solution to tackle the complex and diverse nitrate-containing wastewater in industrial applications. As shown in Supplementary Fig. 11, the electrochemical double-layer capacitance (C_dl) measurement method was employed to estimate the electrochemically active surface area (ECSA) in this study. The ECSA value of V-Cu NAE with abundant defects was five times higher than that of defect-free Cu NW, demonstrating that in situ constructed defects significantly augments the number of active sites. Isotopic labeling experiments and blank comparisons conclusively established that the ammonia originated exclusively from NO₃⁻ rather than external contamination (Supplementary Fig. 12)[35]. Meanwhile, to compare with colorimetric methods using Nessler's reagent (Supplementary Fig. 8), the concentration of ammonium was further determined by ¹H NMR using external standards (Supplementary Fig. 13). The ammonia yields of ¹⁵NH₄⁺-¹⁵N and ¹⁴NH₄⁺-¹⁴N obtained by ¹H NMR were found to be

highly consistent with the quantitative results obtained by colorimetric methods, as demonstrated in Supplementary Fig. 14, thus confirming the accuracy of these quantitative techniques[36].

To investigate the reaction process and mechanism of NO₃RR on the V-Cu NAE, advanced Operando SR-FTIR spectroscopy was conducted. The laboratory setup and the electrolytic cell used for these measurements are depicted in Supplementary Fig. 15 and Supplementary Fig. 16, respectively[37]. To provide a baseline for comparison, control experiments without nitrate were conducted under similar conditions (Supplementary Fig. 17). As shown in Fig. 3a, we collected and analyzed infrared signals ranging from 4000 to 1250 cm⁻¹ under an applied potential between −0.1 and −0.5 V (vs. RHE) and at OCP. In the SR-FTIR operation results of 3600-3000 cm⁻¹ (Fig. 3b), free ammonia is represented at 3185 cm⁻¹, and the peak at 3400 cm⁻¹ is attributed to water. Furthermore, as the voltage decreased from −0.1 to −0.5 V, the peak intensity increased, indicating an increase in NH₃ yield[38].

Within the range of 1700–1250 cm⁻¹ (Fig. 3c), peaks observed at 1582 cm⁻¹ and 1475 cm⁻¹ were ascribed to the NOₓ intermediates and NH₄⁺, respectively[6,25,39]. It is clear that the production rates of NOₓ and NH₄⁺ incrementally rose as the applied potential decreased from −0.1 to −0.5 V (vs. RHE). The scarce NO₂⁻ content at −0.5 V, attributed to the

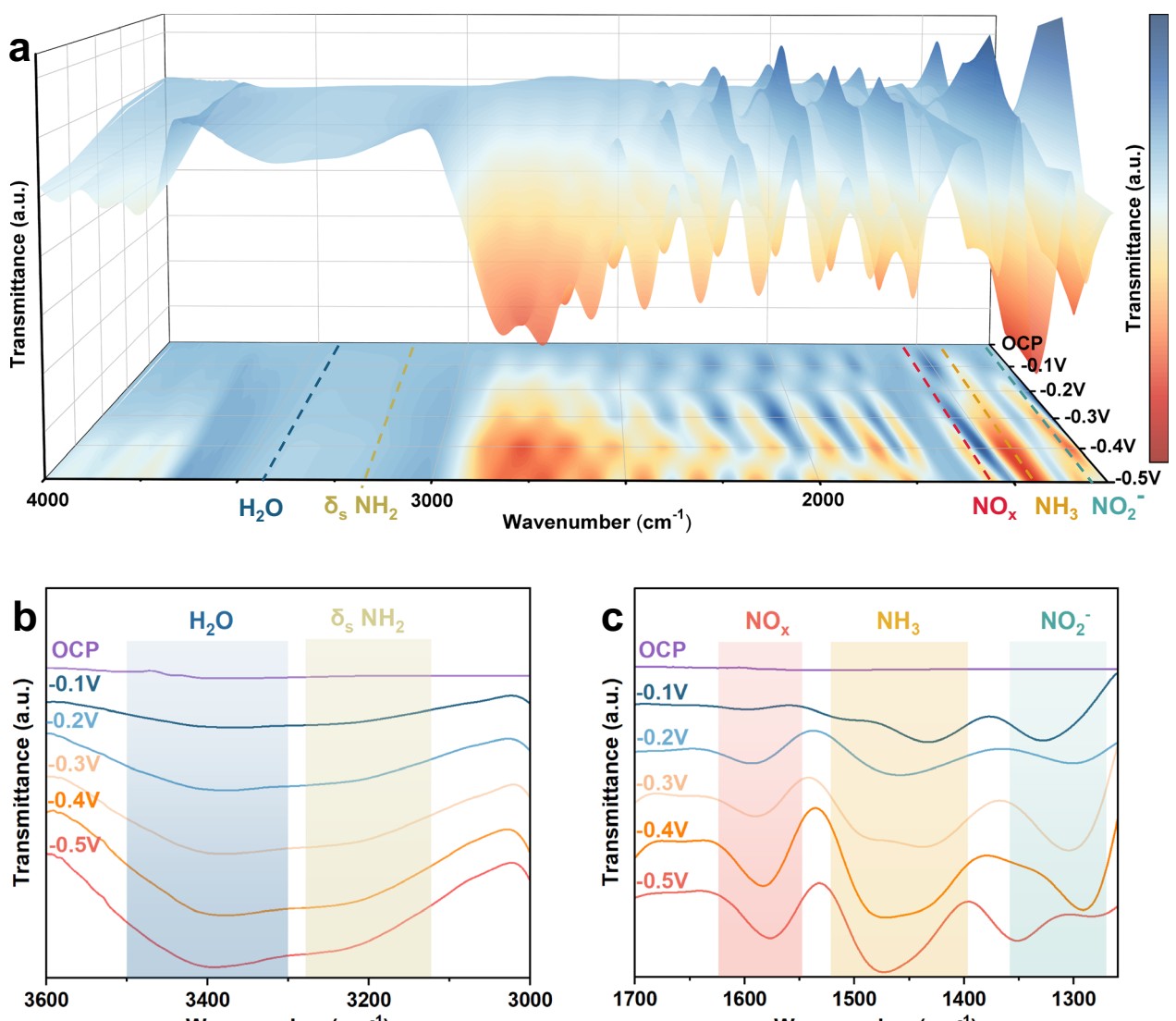

**Fig. 3 | Operando Synchrotron radiation Fourier Transform Infrared Spectroscopy (FTIR) measurements under various potentials for V-Cu NAE during NO₃RR. a** Three-dimensional FTIR spectra and corresponding contour maps in the range of 4000–1250 cm⁻¹, measured under potentials from −0.1 to −0.5 V vs. RHE and at OCP (Open Circuit Potential). Color scale with arrow indicating the change in signal intensity from low to high. **b** Infrared signals in the range of 3600–3000 cm⁻¹. **c** Infrared signals in the range of 1700–1250 cm⁻¹.

1286 cm⁻¹ peak, could be a consequence of its swift depletion at higher voltages[40]. Simultaneously, the three-dimensional FTIR spectra and corresponding contour maps (Fig. 3a) offered a more direct reflection of the intermediate content trend.

To elucidate the superior nitrate removal ability and ammonia production performance of V-Cu NAE, DFT calculations were conducted on Cu(111) facets and defective Cu(111) facets based on earlier observations. Four water molecules were explicitly added to the catalyst surface to account for the stabilizing effect of hydrogen bonding on intermediates, while an implicit solvation model was adopted to treat the solvent environment as a polarizable continuum. The V-Cu model was established by removing a surface Cu atom from the Cu(111) (as depicted in Supplementary Fig. 18 and Supplementary Fig. 19). Additionally, the constant electrode potential simulation and the potential-dependent activation energy simulation were employed to study the potential-dependent NO₃RR, HER, and water splitting energetics.

The band center model served as a robust methodology for examining the binding energy and catalytic interactions between transition metals and reactants or intermediates[41,42]. In the V-Cu model,

Cu atoms near the defect exhibit an unsaturated state, possessing a d-band center closer to the Fermi level (Fig. 4a). This leads the V-Cu(111) to exhibit stronger adsorption for various intermediates compared to Cu(111)[43,44]. Based on experimental findings, the optimum performance of V-Cu(111) was observed at $U = -0.289$ V vs RHE, hence all energies in Fig. 4a–e were calculated at this potential. Firstly, the total energies of intermediates involved in water splitting and HER on Cu(111) and V-Cu(111), such as slab, *H, *OH, and *OH + *H, vary with $U_{SHE}$, as shown in Supplementary Fig. 20. Figure 4b presents the water splitting free energy diagram for Cu(111) and V-Cu(111). The results indicate that V-Cu(111) exhibits stronger water molecule adsorption and has a lower water splitting activation energy (0.38 eV) than Cu(111) (0.72 eV)[45]. This implies that V-Cu is more capable of supplying active protons to facilitate the protonation process in NO₃RR, culminating in a high ammonia production current density (1000 mA·cm⁻²) for the V-Cu NAE[19]. Simultaneously, V-Cu(111) has a higher HER reaction barrier (0.84 eV) compared to Cu(111) (0.64 eV), indicating a stronger suppression ability for HER, and FE was improved.

For NO₃RR on Cu(111) and V-Cu(111), Supplementary Fig. 21 showcases the potential intermediates and a complex adsorption/desorption,

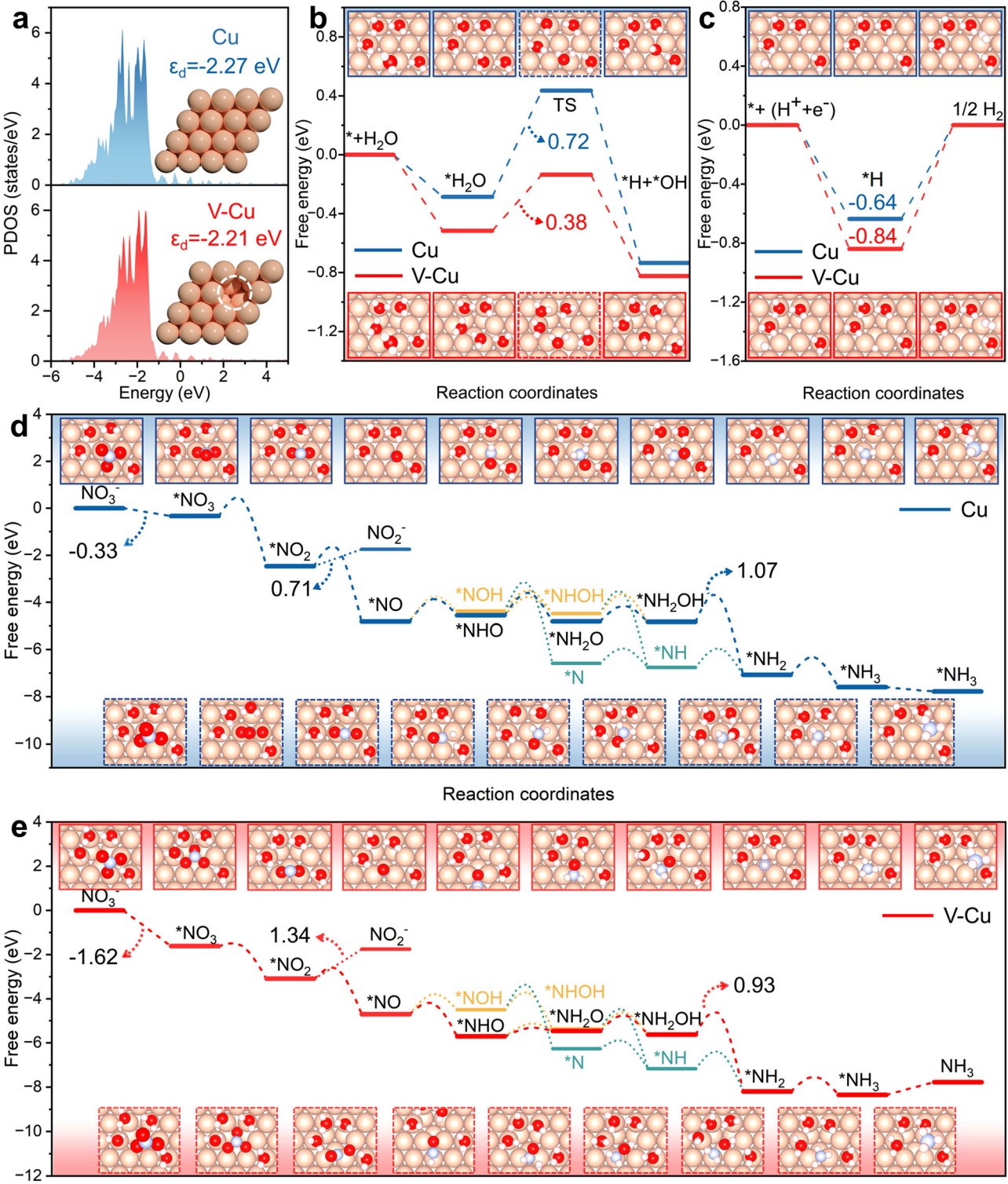

**Fig. 4 | DFT calculations. a** Illustrated are the projected density of states (PDOS) along with structural models for both V-Cu NAE and Cu NWs. **b** Depicted water splitting reaction pathways for both V-Cu and Cu when $U = -0.289$ V vs RHE. **c** Showcases the hydrogen evolution reaction pathways for V-Cu and Cu at $U = -0.289$ V vs RHE. **d** The NO₃RR pathways for Cu are displayed, with the optimal pathway distinctly highlighted in blue at $U = -0.289$ V vs RHE. **e** The NO₃RR pathways for V-Cu are demonstrated with the optimal pathway prominently marked in red at $U = -0.289$ V vs RHE. Importantly, these reaction pathways are complemented with free energy diagrams and corresponding models. Models framed in solid lines denote steady-states, while those with dashed lines signify transition states (TS). For ease of differentiation, models within red frames represent V-Cu, and those within blue frames denote Cu.

deoxygenation, and protonation reaction network. The total energies of intermediates associated with $NO_3RR$ on Cu(111) and V-Cu(111) like slab, *$NO_3$, *$NO_2$, *NO, *NOH, *NHO, *NHOH, *$NH_2O$, *$NH_2OH$, *N, *NH, *$NH_2$, *$NH_3$, *O, *OH vary with $U_{SHE}$, as illustrated in Supplementary Fig. 22. The transition states for the elementary steps of $NO_3RR$ on Cu(111) and V-Cu(111) were identified using the Slow-growth method (Supplementary Fig. 23 and Supplementary Fig. 24). Figure 4d, e outlines the reaction pathways of $NO_3RR$ on Cu(111) and V-Cu(111), encompassing all ground-state free energies and kinetic barriers. Based on the activation energies of each elementary step, the optimal reaction pathways for Cu(111) and V-Cu(111) were ascertained to be $NO_3^-$(l) →*$NO_3$ → *$NO_2$ → *NO → *NHO → *$NH_2O$ → *$NH_2OH$ → *$NH_2$ → *$NH_3$ → $NH_3$(g).

The low-coordinated V-Cu(111) showcases enhanced adsorption capabilities for nitrate and nitrite, with adsorption free energies of −1.62 eV and −1.34 eV, respectively. In comparison to Cu (−0.33 eV and −0.71 eV), V-Cu(111) captures nitrate from the solution more swiftly, while also suppressing nitrite desorption. The rate-determining step for both V-Cu(111) and Cu(111) occurs during the protonation of *$NH_2OH$, with activation barriers of 0.93 eV and 1.07 eV, respectively. This indicates a competitive edge for V-Cu(111) in ammonia conversion.

Furthermore, microkinetic simulations were executed to calculate the yield of nitrite and ammonia for V-Cu(111) and Cu(111) at $U = -0.289$ V vs RHE. The results from Supplementary Fig. 22a demonstrate that, at equilibrium, 60% of the sites on Cu(111) are occupied by *O and *OH, with the remaining sites vacant. Slower protonation rates of oxygen species and weaker adsorption energies are the culprits for this. Due to its more negative nitrate adsorption energy and faster *$NO_3$ deoxygenation rate, sites on V-Cu(111) are rapidly consumed at the onset of the reaction, leading to the prompt deoxygenation of *$NO_3$ to produce *$NO_2$ and *O (Supplementary Fig. 25b). Owing to the faster protonation rates of oxygen species on V-Cu(111), *O and *OH are also rapidly consumed. Once steady-state equilibrium is attained, the surface is completely covered by *$NO_2$, at which point the deoxygenation rate of *$NO_2$ becomes the critical step in converting nitrate to ammonia. Finally, as depicted in Supplementary Fig. 25c, the steady-state yields of nitrite and ammonia for Cu and V-Cu were determined. The results show that the ammonia yield for V-Cu is $1.186*10^{-7}$ mol/s, with nitrite production at $8.879*10^{-6}$ mol/s. In contrast, Cu yields $1.289*10^{-9}$ mol/s of ammonia with a nitrite output of $4.8*10^{-4}$ mol/s. These microkinetic simulation results affirm the higher selectivity of V-Cu(111) for ammonia production.

To further explore the industrialization potential of the catalyst, a two-electrode flow electrolytic cell coupled with $NO_3RR$ and GOR reaction was designed to enhance energy utilization efficiency and wastewater treatment capacity (Fig. 5a). Specifically, the V-Cu NAE and commercial Ni foam were utilized as the cathode and the anode respectively. Actual industrial wastewater ($NO_3^-$-N content of 527 ppm) and 0.1 M glycerin served as the electrolyte for the cathode and anode, respectively. A detailed schematic diagram of the two-electrode flow cell device was presented in Supplementary Fig. 32. The LSV curves depicted in Fig. 5b indicated that the addition of 0.1 M glycerol (GLY) reduced the voltage required to achieve an industrial-grade current density of 500 mA cm$^{-2}$ to just 1.34 V, a substantial drop of 260 mV. To evaluate the wastewater treatment capacity and ammonia production performance of the device, a transformation experiment of 1 L actual industrial wastewater was conducted with 1 cm$^{-2}$ V-Cu NAE. As shown in Fig. 5c, with a low cell voltage of −1.4 V, the $NO_3^-$ concentration diminished progressively during electrolysis, accompanied by a decrease in current density from 600 mA·cm$^{-2}$ to 100 mA·cm$^{-2}$. Following 30 h of electrolysis, only 0.52 ppm of $NO_3^-$-N and 0.38 ppm of $NO_2^-$-N remained, both significantly below the WHO regulations for drinking water, which corresponds to the $NH_3$ selectivity of 99.9%, the $NO_3^-$ conversion of 99.9% and the FE over 80%. It was observed that the

1 cm$^2$ catalyst successfully reduced the $NO_3^-$ and $NO_2^-$ content of 1 L of actual industrial wastewater to the drinking water standard within 30 h, which demonstrates its impressive wastewater treatment capacity. Additionally, the $^1$H NMR results (Fig. 5d) indicated that glycerol was oxidized at the anode into a high value-added formate, with a FE of 81.3% (Supplementary Fig. 33), thereby enhancing the overall economic benefits[46]. Given that stability is a critical parameter for industrialization, a stability test was conducted on the device using 5 L of wastewater at 1.4 V, with periodic updates taken every 30 h. As illustrated in Fig. 5e, the $NH_3$ production rate and FE of the catalyst remained stable even after 100 h, revealing exceptional stability. To further investigate the post-operational stability and durability of the catalyst, we conducted comprehensive characterizations. As shown in Supplementary Fig. 34, the crystalline phase of the catalyst remained consistent as metallic Cu phase after the stability tests. From the SEM examinations, several morphological changes were identified after the cycling tests (Supplementary Fig. 35). Specifically, some nanowire arrays detached from the Cu foam substrate, contrasting with their initial state as shown in Supplementary Fig. 35a, c, e, g. In addition, aggregation of nanowires was observed in certain areas (Supplementary Fig. 35f), and fractures were evident in some nanowires (Supplementary Fig. 35h). These observed morphological changes might contribute to performance variations detected during the cycling tests. Furthermore, XPS analysis indicated that, relative to the pristine sample, the Cu peaks of the cycled V-Cu NAE still displayed a significant shift when juxtaposed with those of the defect-free Cu NWs (Supplementary Fig. 36). This persistent shift corroborates the enduring stability of the Cu vacancies throughout the cycling process, underscoring the sustained presence of Cu vacancies in the catalyst. Our two-electrode device surpassed all previously reported performances in electrocatalytic nitrate reduction to ammonia[47–49].

During the treatment of authentic industrial wastewater, air stripping has been effectively utilized to extract ammonia ($NH_3$), subsequently yielding a valuable end product[50,51]. Figure 6a illustrated that under operational conditions of 70 °C and an air flow, approximately 98% of the $NH_3$ could be effectively removed from the treated wastewater without affecting other impurities. Following the extraction of $NH_3$, hydrochloric acid was utilized to capture the ammonia, yielding ammonium chloride ($NH_4Cl$). Subsequently, a rotary evaporator facilitated the separation of $NH_4Cl$. Additionally, the direct infusion of $NH_3$ into water generates ammonia monohydrate ($NH_3 \cdot H_2O$). Both procedures remarkably contributed to the conversion of over 80% of the $NH_3$ present in the wastewater into chemical products, as depicted in Fig. 6b. X-ray diffraction (XRD) and proton nuclear magnetic resonance ($^1$H NMR) analyses were performed to verify the purity of the resultant products. As evidenced by Fig. 6c, d, these analytical techniques confirmed the high-purity $NH_4Cl$(s) and $NH_3 \cdot H_2O$(aq) obtained from the treated industrial wastewater, with no noticeable impurities. To conclusion, the implemented air stripping method proves to be an effective and selective strategy for extracting $NH_3$ from real industrial wastewater, thus yielding high purity valuable end products such as $NH_4Cl$ and $NH_3 \cdot H_2O$. This study set the foundation for further refinement and optimization of wastewater treatment processes, ultimately contributing towards a more sustainable and efficient utilization of resources.

In summary, we successfully synthesized a defect-rich Cu NAE obtained by in-situ electrochemical reduction of $Cu_3N$ NWs. Operando Synchrotron radiation-FTIR and DFT calculations revealed that the defect induced triple synergistic modulation: enhanced nitrate adsorption, promoted water dissociation, and suppressed hydrogen evolution. Consequently, the V-Cu NAE catalyst attains 50–1100 mA cm$^{-2}$ within a nitrate concentration range of 1–100 mM, maintaining a FE exceeding 90% and effectively reducing nitrate levels to comply with the WHO drinking water standards. Furthermore, we explored the industrial potential of the V-Cu NAE by using a two-

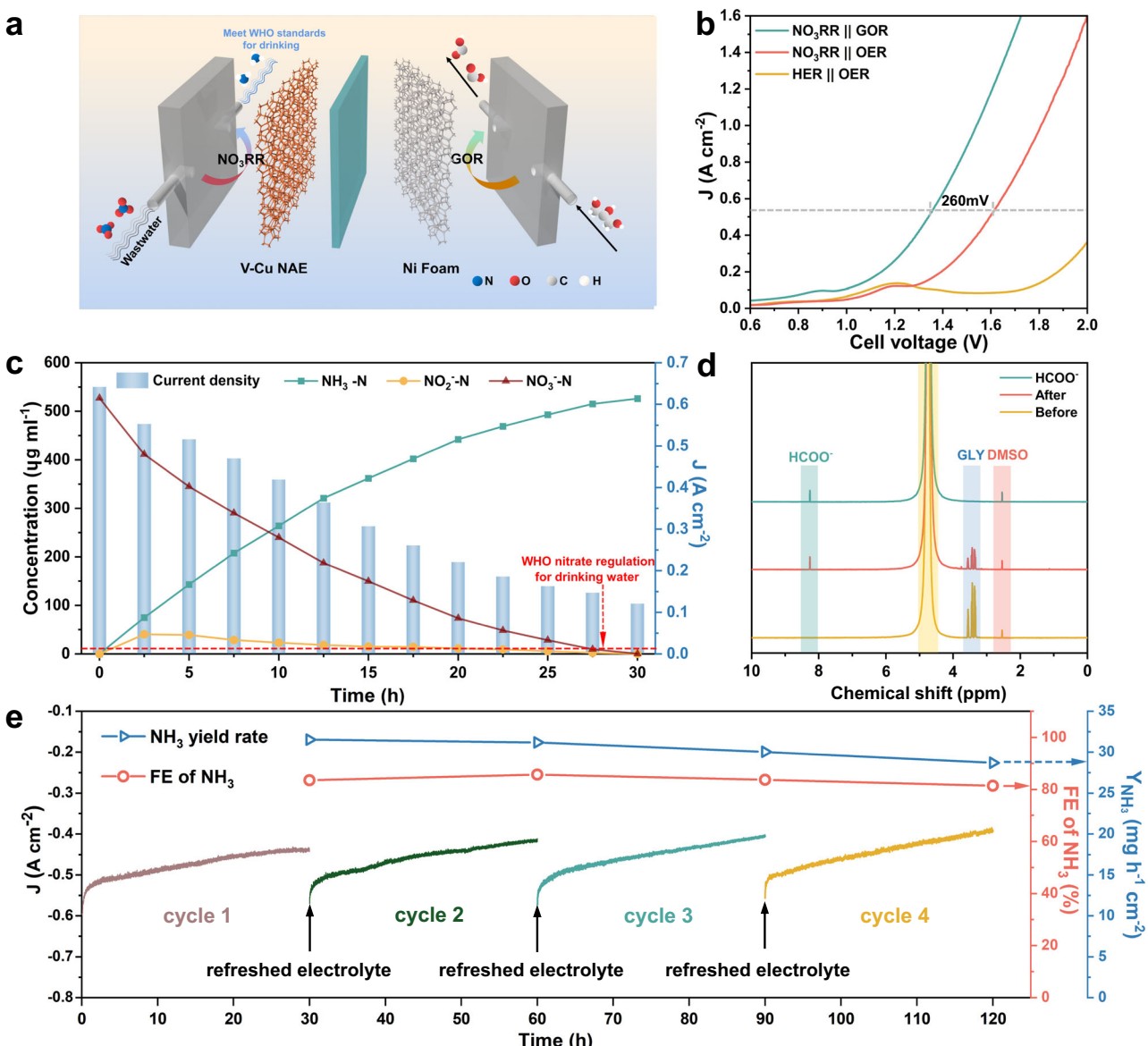

**Fig. 5 | Electrochemical NO₃RR with actual industrial wastewater coupled with Glycerol Oxidation Reaction in a two-electrode system. a** Scheme of the electrolyzer coupling NO₃RR with industrial wastewater and Glycerol Oxidation Reaction (GOR). **b** LSV curves (without IR compensation) using the electrolyzer with and without glycerol at the anode. The measured resistance of the two-electrode system was $0.35 \pm 0.02\ \Omega$. **c** Complete nitrate removal using V-Cu NAE (1 cm²) with 1 L of actual industrial wastewater (containing 0.5 M K₂SO₄ with 527 ppm NO₃⁻-N) at

1.4 V. After 30 h of electrolysis, only 0.52 ppm NO₃⁻-N and 0.38 ppm NO₂⁻-N remained, both significantly below the WHO regulations for drinking water (NO₃⁻-N < 11.3 ppm and NO₂⁻-N < 0.91 ppm), with 99.9% NH₃ selectivity, 99.9% NO₃⁻ conversion, NH₃ FE over 80%. The blue data shows the change of current over time. **d** ¹H NMR results for GOR production characterization. **e** Chronoamperometric curves, FE and NH₃ yield rate of V-Cu NAE at 1.4 V by 4 cycles, with each cycle presenting 30 h of long-term electrolysis.

electrode system to treat actual wastewater with 99.9% NH₃ selectivity, 99.9% NO₃⁻ conversion. Our work offers a potential solution for addressing the complex and diverse nitrate-containing wastewater encountered in industrial applications.

## Methods

### Materials characterization

The morphology and structure of the samples were characterized using various techniques, including field-emission scanning electron microscopy (FESEM, SU-8200), transmission electron microscopy (TEM, HIACHI HT7700; Talox F200X), powder X-ray diffractor (XRD, TTR-III), and Raman spectrometer (Renishaw inVia) with a 532 nm excitation laser. High-resolution TEM (HRTEM) images and energy dispersive spectroscopy (EDS) mapping images were recorded using a Talox F200X transmission electron microscope operating at 200 kV.

The chemical state of the samples was analyzed using an ESCALAB 250 X-ray photoelectron spectrometer with an X-ray source (Al Kα, hν = 1486.6 eV). All peaks were calibrated with the C 1s spectrum at a binding energy of 284.8 eV. Ultraviolet-visible (UV-Vis) absorbance spectra were measured using a Shimadzu UV-3600 spectrophotometer. Isotope labeling experiments were conducted and measured using 1H NMR spectroscopy (Bruker 400-MHz system).

**Pre-treatment of Cu foam.** The Cu foam was cut into rectangle with an area of 4 cm × 1 cm, and then ultrasonicated in 1 M HCl aqueous solution, water, respectively.

**Synthesis of Cu(OH)₂ nanowire arrays (Cu(OH)₂ NWs).** The above Cu foam was immersed in a solution of 10 mL of ammonium persulfate (0.285 g) and 10 mL of sodium hydroxide (1.0 g) for 30 min, during

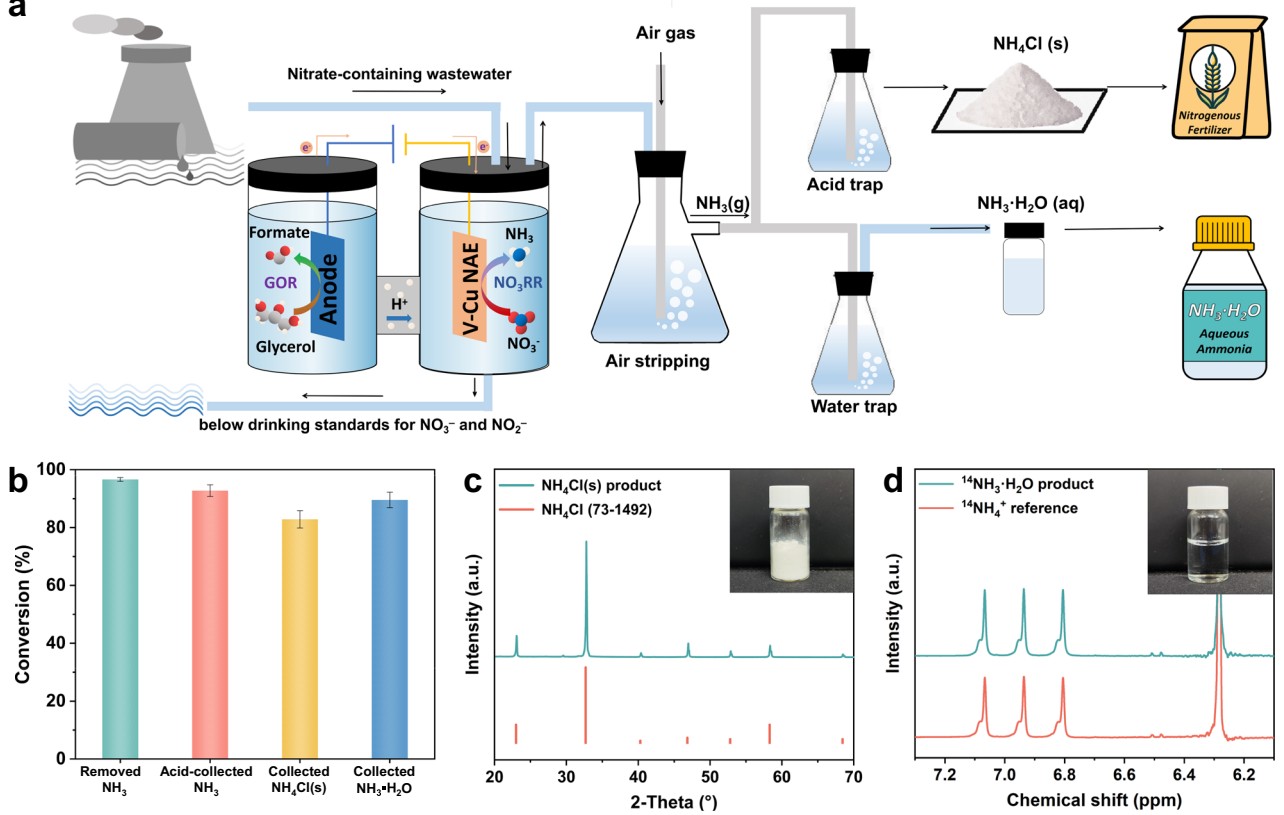

**Fig. 6 | Synthesis and extraction of ammonia products from industrial wastewater. a** Schematic of the ammonia product synthesis process from industrial wastewater to NH$_4$Cl(s) and NH$_3$·H$_2$O(aq). **b** Conversion efficiency of different steps for the ammonia product synthesis process. **c** XRD and picture of synthesized NH$_4$Cl(s) products. **d** $^1$H NMR and picture of the synthesized NH$_3$·H$_2$O(aq). Error bars represent the standard deviations calculated from three independent measurements.

which the surface of the Cu foam changed color to cyan and the solution became blue. After that, the cyan-colored Cu foam were washed with deionized water and ethanol, and dried under vacuum at 50 °C.

**Synthesis of Cu$_3$N nanowire arrays (Cu$_3$N NWs).** The Cu(OH)$_2$ NWs were heated at 300 °C for 2 h with the heating rate of 2 °C/min under the NH$_3$ atmosphere. Then, they were cooled down to natural temperature.

**Synthesis of Cu nanowire arrays (Cu NWs).** The Cu(OH)$_2$ NWAs were heated at 300 °C for 2 h with the heating rate of 2 °C/min under the H$_2$/Ar (5:95) atmosphere.

**Synthesis of Cu nanowire arrays rich in vacancy (V-Cu NAE).** The V-Cu NAEs used for nitrate reduction was then produced by in-situ electrochemical reduction of the Cu$_3$N NWs, which removed nitrogen from the surface, thereby constructing V-Cu NAE. CV technology was employed from −0.4 to −1.8 V at a scan rate of 200 mV s$^{-1}$ for at least 20 min until the baseline kept steady. During in situ electrochemical reduction, Cu$_3$N NWs gradually loses nitrogen and becomes Cu nanowire arrays rich in vacancy, thus changing from black to reddish-brown.

### Electrochemical measurements

All the electrochemical measurements of the samples were carried out using a CS Electrochemical Workstation (CS150M, China) in a H-type electrolytic cell separated by a membrane. The copper-based sample on Cu mesh, saturated calomel electrode (SCE) and platinum foil was used as the working electrode, reference electrode and counter electrode, respectively. The surface area of the working electrode was controlled at 1 cm$^2$, and the catalyst loading was 3 mg cm$^{-2}$. 0.5 M K$_2$SO$_4$ solution (35 mL) was evenly distributed to the cathode and anode compartment. KNO$_3$ was added into the cathode compartment for NO$_3^-$ reduction (containing 1, 2, 5, 10, 20, 30, 40, 50, 100, 500, 1000 mM nitrate-N, respectively). Before nitrate electroreduction test, Linear sweep voltammetry (LSV) curves are performed until that the polarization curves achieve steady-state ones at a rate of 10 mV s$^{-1}$ from 0.2 V to −0.6 V vs RHE. Then, the potentiostatic test was carried out at different potentials for 1 h with a stirring rate of 300 rpm. The resistance of the electrochemical cell was determined using potentiostatic electrochemical impedance spectroscopy, conducted over a frequency range from 0.1 Hz to 200 kHz with a signal amplitude set at 10 mV$_{pp}$. Cyclic voltammetry (CV) curves in electrochemical double-layer capacitance (C$_{dl}$) determination were measured in a potential window nearly without the Faradaic process at different scan rates of 20, 40, 60, 80 and 100 mV s$^{-1}$ in a solution containing 200 ppm nitrate-N and 0.5 M K$_2$SO$_4$. The plot of current density at set potential against scan rate has a linear relationship and its slope is the C$_{dl}$.

### Determination of ion concentration

The ultraviolet-visible (UV-Vis) spectrophotometry technique was employed for the quantification of ion concentrations within specific calibration ranges. Below are the adapted methodologies for the detection of various nitrogenous compounds:

**Nitrate-N determination.** Initially, a predefined volume of the sample was diluted to 5 mL within the detection limit. To this solution, 0.1 mL

of 1 M hydrochloric acid and 0.01 mL of a 0.8 wt% solution of sulfamic acid were added. The mixture was then allowed to equilibrate for 20 min under room temperature. An ultraviolet-visible spectro-photometer was used to measure the absorption spectrum, focusing on the absorption peaks at 220 nm and 275 nm wavelengths. The net absorbance was determined using the equation: $A = A_{220nm} - 2A_{275nm}$. A calibration curve was established using a range of standard potassium nitrate solutions to correlate absorbance with concentration, facilitating the determination of nitrate concentrations in test samples.

**Nitrite-N determination.** The colorimetric reagent was prepared by combining p-aminobenzenesulfonamide (4 g), N-(1-Naphthyl)ethylenediamine dihydrochloride (0.2 g), ultrapure water (50 mL), and phosphoric acid (10 mL with a density of 1.70 g/mL). A specific volume of the sample was diluted to 5 mL to fall within the detection limit, followed by the addition of 0.1 mL of the prepared color reagent. The solution was homogenized and allowed to rest for 20 min, after which the absorption intensity at 540 nm was measured. A series of standard sodium nitrite solutions were used to calibrate the concentration-absorbance curve, enabling the calculation of nitrite concentrations in samples.

**Ammonia-N determination.** The detection of ammonia utilized Nessler's reagent as the colorimetric agent. The sample was diluted to a 5 mL volume within the detection range, to which 0.1 mL of a potassium sodium tartrate solution (density = 500 g/L) was added and mixed well. Subsequently, 0.1 mL of Nessler's reagent was introduced, and the solution was uniformly mixed. The absorbance was recorded at a wavelength of 420 nm after a 20-min stabilization period. Calibration was performed using a series of standard ammonium chloride solutions, facilitating the determination of ammonia concentrations.

## Calculation of the yield, conversion rate, selectivity, and Faradaic efficiency

The yield of $NH_3$ was determined using below equation:

$$Yield_{NH_3} = (c_{NH_3} \times V)/(M_{NH_3} \times t \times S) \tag{1}$$

where $c_{NH_3}$ signifies the mass concentration of ammonia in aqueous solution, mol cm$^{-3}$; $V$ specifies the total volume of electrolyte present within the cathode compartment, mL; $M_{NH_3}$ is the molar mass of $NH_3$; $t$ represents the duration of the electrolysis process, h; $S$ is the area of the cathode, cm$^{-2}$;

The conversion rate was calculated using below equation:

$$Conversion = \Delta c_{NO_3^-}/c_0 \times 100\% \tag{2}$$

where $\Delta c_{NO_3^-}$ is the variation in nitrate concentration before and after electrolysis, mol cm$^{-3}$; $c_0$ represents the initial concentration of nitrate, mol cm$^{-3}$.

The selectivity of ammonia or nitrite was obtained using below equation:

$$Selectivity = c/\Delta c_{NO_3^-} \times 100\% \tag{3}$$

where $c$ denotes the generated concentration of ammonia or nitrite, mol cm$^{-3}$.

Faradaic efficiency was assessed through Eq. (4), defined as the ratio of the electric charge utilized for ammonia synthesis to the total charge transferred across the electrode:

$$Faradaic\ efficiency = (8 \times F \times c_{NH_3} \times V)/(M_{NH_3} \times Q) \tag{4}$$

where $F$ is the Faraday constant, valued at 96,485 Coulombs per mole (C/mol); $Q$ is the total amount of charge consumed, C.

## Isotope labeling experiment

Using 99 atom % $K^{15}NO_3$ as feed nitrogen source, isotope labeled nitrate reduction experiment was carried out to determine the source of ammonia. Potassium sulfate of 0.5 M was used as electrolyte, and $K^{15}NO3$ with concentration of 200 ppm and $^{15}NO_3^- - ^{15}N$ was added into the cathode chamber as reactant. After electric reduction, the electrolyte of $^{15}NH_4^+ - ^{15}N$ was taken out. The pH value was adjusted to weak acid by 4 M sulfuric acid, and further quantified by $^1H$ NMR (400 MHz) using maleic acid standard. The calibration curve was established as follows: Firstly, $^{15}N$ with known concentration (50, 100, 150, 200, 250,) was prepared in 0.5 M potassium sulfate ($^{15}N$ solution was used as the standard); secondly, $^{15}NH_4^+ - ^{15}N$ standard solution with different concentration was mixed with 0.02 g maleic acid; Thirdly, 50 μL deuterium oxide ($D_2O$) was added to 0.5 mL of mixed solution for NMR detection. Finally, the peak area ratio of $^{15}NH_4^+ - ^{15}N$ and maleic acid was used for calibration, because the concentration of $^{15}NH_4^+ - ^{15}N$ was positively correlated with the area ratio. Similarly, the amount of $^{14}NH_4^+ - ^{14}N$ was quantitatively measured.

## In-situ Raman spectroscopy

Raman spectroscopic analysis in-situ utilized a Confocal Raman Microscope (Renishaw inVia) equipped with an Olympus 50X (0.5 N.A.) long working distance lens. A 532 nm wavelength laser, with a power output of 2.5 mW and utilizing an 1800 lines/mm grating, was deployed. The $Cu_3N$ NWs measuring 1 cm by 1 cm served as the substrate for the working electrode, while Pt wire and Ag/AgCl were employed as the counter and reference electrodes, respectively. The time-dependent Raman spectroscopy measurement towards electrocatalysis process were under programmed applied potentials in 0.5 M $K_2SO_4$. The electrocatalyst is held for 300 s at the applied potential to reach steady state conditions before recording each spectrum. Each spectrum was acquired over 15 sweeps, spanning the range of 175 to 1900 cm$^{-1}$.

## Operando SR-FTIR measurements

At the National Synchrotron on Radiation Laboratory's BL01B infrared beamline, in-situ SR-FTIR analyses utilized a custom-built reflection apparatus with a ZnSe crystal window for infrared transmission, which allows a wavelength pass-through from 20,000 to 440 cm$^{-1}$ (0.5–23 μm) with a transmittance greater than 68%. The setup included an FTIR spectrometer featuring a KBr beam splitter and multiple detectors, notably a liquid-nitrogen-cooled mercury cadmium telluride detector, integrated with a Bruker Hyperion 2000 infrared microscope and a ×15 magnification objective. A micrometer-thick gap was maintained between the catalyst electrode and the ZnSe window. Initially, the sample was located using a reflection observation mode to focus on the catalyst surface. Measurements were then conducted in a reflection mode with vertical incidence of infrared light. With the light source beam current >400 mA, each infrared absorption spectrum was acquired by averaging 128 scans at a resolution of 4 cm$^{-1}$. Before measurement, the catalyst electrode's background spectrum was collected at open-circuit voltage (OCP). The electrocoupling reaction potentials were systematically varied from −0.4 to −0.9 V versus RHE, in steps of 0.1 V. It's important to note that the instrument automatically subtracted the background obtained at OCP from the test results.

## Computational detail

The Vienna ab initio simulation package (VASP) based on density functional theory (DFT) was utilized for all calculations in this study[1]. We adopted projector augmented wave (PAW) pseudopotentials and a semi-local generalized gradient approximation (GGA) with Perdew-Burker-Ernzerhof (PBE) exchange-correlation energy[2]. To reasonably account for weak, long-distance van der Waals (vdWs) effects, we employed an empirical dispersion-corrected DFT method (DFT-D3)[3]. The planewave expansion kinetic energy cutoff was set to 400 eV, and the self-consistent field (SCF) iteration convergence threshold was set

to $10^{-5}$ eV. Geometry optimization using the conjugate gradient method was carried out, with forces on each atom less than 0.03 eV/Å.

## Adsorption energies of nitrate and nitrite

For DFT, obtaining the energy of the nitrate ion is challenging because DFT cannot accurately obtain the energy of charged molecules. We can derive the adsorption energy of the nitrate using thermodynamic data from the CRC Handbook combined with the accurate energy that DFT can calculate. In fact, this is how we handled it in the manuscript we initially submitted, but we did not detail this aspect in the supporting information or main text. These details will be re-included in the computational details section of the supporting information. The specific derivation process is as follows:

$$NO_3^-(l) + (H^+ + e^-) + * \rightarrow HNO_3(l) + * \quad (S1)$$

$$HNO_3(l) \rightarrow HNO_3(g) \quad (S2)$$

$$HNO_3(g) + * \rightarrow *NO_3 + (H^+ + e^-) \quad (S3)$$

$$NO_3^-(l) + H^+ + * \rightarrow *NO_3 + (H^+ + e^-) \quad (S4)$$

According to Hess's law, S4 represents the adsorption process of the nitrate ion, and the Gibbs free energy of S4 can be derived from S1, S2, and S3, namely:

$$\Delta G_{S4} = \Delta G(NO_3) = \Delta G_{S1} + \Delta G_{S2} + \Delta G_{S3} \quad (S5)$$

Where $\Delta G_{S1}$ and $\Delta G_{S2}$ can be obtained by referring to the CRC Handbook of Chemistry and Physics, with values of 0.317 eV and 0.075 eV, respectively. $\Delta G_{S3}$ can be derived from DFT calculations.

$$\Delta G_{S3} = \Delta G_{ads}(*NO_3) = G(*NO_3) + \mu(H^+) + \mu(e^-) - G(HNO_3(g)) - G(*) \quad (S6)$$

Where $G(*NO_3)$, $G(HNO_3(g))$ and $G(*)$ are the free energies of the system after $NO_3$ adsorption on the slab, the free energy of the gaseous $HNO_3$ molecule, and the free energy of the slab, respectively. $\mu(H^+)$ and $\mu(e^-)$ represent the chemical potentials of $H^+$ and $e^-$, respectively. The calculation of $\Delta G_{S3}$ involves the chemical potentials of protons and electrons. Therefore, even when the energy of $HNO_3$ is used as a reference for calculating nitrate adsorption, this process remains potential-dependent.

Additionally, to consider the desorption of nitrite, the energy of nitrite is also required. Similarly, the energy of nitrite is derived using thermodynamic data from the CRC Handbook combined with DFT for accurate energy calculation.

$$NO_2^-(l) + (H^+ + e^-) + * \rightarrow HNO_2(l) + * \quad (S7)$$

$$HNO_2(l) \rightarrow HNO_2(g) \quad (S8)$$

$$HNO_2(g) + * \rightarrow *NO_2 + (H^+ + e^-) \quad (S9)$$

$$NO_2^-(l) + H^+ + * \rightarrow (H^+ + e^-) \quad (S10)$$

According to Hess's law, the Gibbs free energy of S7 can be derived from S8, S9, and S10, namely:

$$\Delta G_{S7} = \Delta G(NO_2) = \Delta G_{S8} + \Delta G_{S9} + \Delta G_{S10} \quad (S11)$$

Where $\Delta G_{S8}$ and $\Delta G_{S9}$ can be obtained by referring to the CRC Handbook of Chemistry and Physics, with values of −0.198 eV and 0.069 eV, respectively. $\Delta G_{S10}$ can be derived from DFT calculations.

$$\Delta G_{S10} = \Delta G_{ads}(*NO_2) = G(*NO_2) + \mu(H^+) + \mu(e^-) - G(HNO_2(g)) - G(*) \quad (S12)$$

Where $G(*NO_2)$, $G(HNO_2(g))$ and $G(*)$ are the free energies of the system after $NO_3$ adsorption on the slab, the free energy of the gaseous $HNO_3$ molecule, and the free energy of the slab, respectively. $\mu(H^+)$ and $\mu(e^-)$ represent the chemical potentials of $H^+$ and $e^-$, respectively.

The free energy consists of three terms, as shown in Eq. (S13), which are the DFT computed energy ($E_{DFT}$), zero-point energy ($ZPE$), and the contribution from entropy change ($TS$):

$$G = E_{DFT} + ZPE - TS \quad (S13)$$

In this case, the temperature $T$ is taken as 298.15 K.

## Potential dependent reaction energy

Nørskov's computational hydrogen electrode (CHE) method is utilized to describe the chemical potential of protons and electrons. Specifically, using the linear free energy dependency on electronic energy under this potential, the free energy dependency of the proton-electron pair on the electrode potential is determined, that is, the electronic energy is offset by -e$U$.

$$\mu(H^+) + \mu(e^-) = \frac{1}{2}\mu(H_2) - eU \quad (S14)$$

Here, $e$ represents the elementary positive charge ($1.602176634 \times 10^{-19}$ C), and $U$ is the electrode potential relative to Reversible Hydrogen Electrode (RHE). For the protonation process $i$, its free energy barrier $\Delta G_i$ will undergo the following changes:

$$\Delta G_i(U) = \Delta G_i(U = 0 \, vs. RHE) + eU \quad (S15)$$

Where, $U$ is the electrode potential relative to the RHE. Under a more negative reduction voltage, the protonation process becomes increasingly favorable.

For the adsorption process of the nitrate ion and the desorption process of the nitrite ion, their free energy barriers will undergo the following changes:

$$\Delta G(NO_3) = \Delta G(NO_3) - eU \quad (S16)$$

$$\Delta G(NO_2) = \Delta G(NO_2) - eU \quad (S17)$$

Under a more negative reduction voltage, the adsorption process of the nitrate ion becomes increasingly difficult, while the desorption process of the nitrite ion becomes increasingly favorable. At the same time, the protonation process becomes relatively easier.

## Constant potential simulation

To more precisely account for actual reaction conditions, such as pH and electrode potential, we adopted the double-referencing method developed by Duan et al. to study the energetics of $NO_3RR$. This approach elucidates the intrinsic catalytic active sites and pH-dependent activity of $NO_3RR$ under actual reaction conditions. The implicit solvation model is implemented through VASPsol. A relative dielectric constant of 80 was set to simulate the water electrolyte. VASPsol assigns an effective surface tension parameter value of 0, thereby neglecting the contribution of cavitation energy. The compensating charge was simulated by a linearized Poisson-Boltzmann

model with a Debye length of 3.0 Å. To clarify the reaction mechanism under different electrode potentials, we adjusted varying amounts of excess charge (q), varying from −2.0 e to +2.0 e in increments of +0.5 e.

The free energy under the potential-dependent grand canonical ensemble is as follows:

$$E_{free}(U) = E_{DFT} + \int_0^q |\overline{V_{tot}}| dQ + qW_f/e \qquad (S18)$$

Where $E_{free}$ represents the free energy under the grand canonical ensemble, $E_{DFT}$ is the energy derived from DFT calculations, $|\overline{V_{tot}}|$ is the average electrostatic potential of the model, $W_f$ is the work function of the charged system, and $q$ is the number of electrons doped into the system.

The electrode potential relative to the standard hydrogen electrode (SHE) can be determined from the system's work function $W_f$.

$$U_{SHE} = W_f/e - 4.6V \qquad (S19)$$

Where 4.6 V is the absolute electrode potential reference of the standard hydrogen electrode (SHE) in VASPsol.

A quadratic function relationship exists between $E_{free}$ and $U_{SHE}$

$$E_{free}(U_{SHE}) = -\frac{1}{2}C(U_{SHE} - U_{PZC})^2 + E_{PZC} = aU_{SHE}^2 + bU_{SHE} + c \quad (S20)$$

Where $C$ represents the capacitance of the system. $U_{PZC}$ and $E_{PZC}$ are the electrode potential and free energy, respectively, relative to SHE in a zero-charge system. The vertex form of the quadratic function relationship can be simplified to the standard form of a quadratic function, where $a$, $b$, and $c$ are parameters that need to be fitted. After obtaining $E_{free}$ under actual reaction conditions, $E_{DFT}$ in Eq. (S13) can be replaced with $E_{free}$ to simulate the free energy under constant potential.

### Potential dependent activation energy barrier
Akhade et al. have devised a method for addressing activation energies that vary with potential.

For non-electrochemical surface protonation:

$$*A + *H \rightleftharpoons *AH + * \qquad (S21)$$

Utilizing DFT to calculate the minimal energy geometry and hydrogenation transition state of reactants allows the determination of activation energy ($E_a$) for processes outside of electrochemical surface protonation. For the reaction $*A + H^+ + e^- \rightleftharpoons AH*$, the activation energy, represented as $E_a(U^0)$, is equivalent to Ea. At $U_0$, the equilibrium potential, the state $\mu(*H)$ that is non-electrochemically analogous finds its balance with the electrochemical counterpart, $\mu(H^+(aq) + e^-)$. In this context, $U^0$ corresponds to the free energy of hydrogen adsorption ($\Delta G_H$) at a given surface measured at 0 V versus RHE. One can calculate the free energy variation for electrochemical hydrogenation on the surface using the formula $U^0 = [G(*A + *H) - G(*A) - 0.5*G(H_2)]/e$. The forward activation energy at potential U is presumed to adhere to the Butler-Volmer formalism, taking into account the transfer of n electrons:

$$E_a(U) = E_a(U^0) + \alpha ne(U - U^0) \qquad (S22)$$

In this study, we fixed α at 0.5 across all basic steps, a choice frequently validated as a practical approximation.

### Microkinetic Simulations
Our investigation employed comprehensive microkinetic simulations to dissect the surface reaction mechanisms implicated in nitrate

reduction processes. Initially, we ascertained the forward and reverse rate constants for the elementary reactions involved, derived via the classical Arrhenius equation. Furthermore, the adsorption rates for non-activated molecular adsorption were computed following the Hertz-Knudsen equation, taking into account the collision rates of gas-phase molecules with the surface.

The MKMXXC microkinetic modeling software suite was utilized to simulate the nitrate reduction rates under standard conditions, specifically at a reaction temperature of 300 K and a molar ratio of nitrate to proton of 1:1. For each reaction step considered in our model, a corresponding differential equation was crafted, incorporating the partial pressures of reactants and the coverage of surface species.

To achieve steady-state surface coverages, these differential equations were integrated over an extended duration until the variations in surface coverage were exceedingly minor (less than $10^{-12}$), thereby ensuring the precision of our simulations. Subsequently, the reaction rates for each elementary step were calculated based on these steady-state surface coverages.

This rigorous approach allowed us to construct an intricate microkinetic model of the nitrate reduction reaction, detailing each elementary step and their contributions to the overall reaction pathway. Moreover, it provided a microscopic insight into the mechanistic underpinnings of the nitrate reduction process.

### Slow-growth method
The free energy barriers in the simulations were computed using both Ab Initio Molecular Dynamics (AIMD) and the Slow-growth method within the canonical (NVT) ensemble, regulated by Nosé-Hoover thermostats at a fixed temperature of 300 K and a time step of 1.0 fs. The Slow-growth method hinges on the geometric parameter λ, a collective variable (CV), which is varied from an initial state (λ0) to a final state (λ1) with a specified velocity of transformation (ν). The Helmholtz free energy difference (ΔF) is calculated by integrating the potential energy's derivative with respect to λ over the simulation duration, which, in the limit of infinitesimally small ν, corresponds to the Gibbs free energy difference (ΔG) between the initial and final states. The detailed parameters for this method applied to various elementary reactions are included in the Supplementary Materials.

A thermodynamic cycle was utilized to calculate the Gibbs free energy of adsorption for $NO_2$ and $NO_3$ in the gas phase. The relevant thermodynamic values were sourced from the CRC Handbook of Chemistry and Physics. The cycle involves the calculation of Gibbs free energy differences for the adsorption processes of $NO_2$ and $NO_3$, with the respective energy values explicitly indicated in the provided schemes.

## Data availability
The raw data of the figures in the main manuscript are available in figshare with the identifier(s) https://doi.org/10.6084/m9.figshare.25124129. All other data needed to evaluate the conclusions in the paper are present in the paper and the Supplementary Information or can be obtained from the corresponding authors upon request.

## Code availability
The code used in this work can be obtained from the corresponding authors on request.

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

## Acknowledgements

G. Q. Z. acknowledges the National Natural Science Foundation of China (Grant No. 52072359), the Recruitment Program of Global Experts and the Fundamental Research Funds for the Central Universities (WK2060000016). The numerical calculations in this paper have been done in the Supercomputing Center of University of Science and Technology of China. The authors are grateful to infrared beamline (BL01B) at National Synchrotron Radiation Laboratory for the experimental beamtime support. This work was partially carried out at the Instruments Center for Physical Science, University of Science and Technology of China.

## Author contributions

B.C.Z., Z.C.D., and G.Q.Z. contributed to the conception of the study. B.C.Z. performed the experiments. Y.X.C. conducted the Density Functional Theory (DFT) calculations analysis. M.Y.C. provided assistance with the Scanning Electron Microscope (SEM) analysis, and H.K.Z. assisted with the electrochemical in situ Fourier Transform Infrared (FTIR) spectroscopy analysis. P.Y.F., K.B.Q., and Y.Y.Z. provided help in the experimental part. G.Q.Z. contributed significantly to the analysis and manuscript preparation. The project was supervised by G.Q.Z. All authors participated in the analysis with constructive discussions.

## Competing interests

The authors declare no competing interests.
