## [Peer Review File · Nature Communications]

REVIEWER COMMENTS

Reviewer #1 (Remarks to the Author):

Zhang et al. reported a combined experimental and theoretical work to study the electrochemical ammonia synthesis via nitrate electroreduction. The defect-rich Cu nanowire array electrode (V-Cu NAE) was generated by in-situ electrochemical reduction. Operando Synchrotron radiation-FTIR and density functional theory (DFT) calculations were used to explain these issues. It claims there are synergy over V-Cu NAE to simultaneously enhance NO₃RR activity, suppressing HER, and promoted water dissociation. Overall, there are two important achievements including the activity of ammonia production at varying nitrate concentration and the extraction of ammonia products from industrial wastewater. However, there are plenty of fundamental issues unclear.

1. The authors chose to refer to the HNO₃ molecule in studying the first step of nitrate conversion. As the nitrate is negatively charged, it will become more and more difficult at reducing potentials. In other words, the adsorption energies of nitrate will be potential dependent and become more and more positive (at more reducing potentials). Instead, the adsorption of HNO₃ should be more favorable and independent on electrode potentials. The authors should carefully check the literature and clarify how the nitrate was converted at the first step.

2. It is obviously incorrect to describe the hydrogen evolution by water dissociation on the surface because it is a chemical process without electron-proton transfer between reaction plane and electrode. At the reducing potentials, the electrochemical protonation by either hydronium or water should be faster. Based on the proposed mechanism shown in Fig. 4b, the HER performance should be hard to be affected by electrode potentials, while it is against lots of experiments. BTW, what is the pH value at the electrochemical interface? Please also convert the SCE to RHE scale and be more friendly for readers.

3. Why the proton has to be transferred to O*, instead of via NHO*, at the first protonation? Why NH₂OH* was considered in the mechanism study? Why the NH₃* was not desorbed to complete a catalytic cycle? The whole process is potential dependent of ammonia production, what is the potentials at Fig. 4d. How the chemical potential of proton and electron were calculated for these equations listed in S1-S8?

4. Besides the chemical potential of electron and protons, the adsorption energies of species can also be affected by electric fields. How much is the electric field effects on the adsorption energies shown in Fig 4.

5. If the d-band model and the description of d-band center is still valid in electrified metals? Why?

6. The high selectivity of ammonia is the major achievements in this work. It was observed in the previous works it is quite difficult to achieve high ammonia selectivity at low overpotentials. However, the selectivity between nitrite and ammonia productions was not explained well. We need all kinetic barriers in fig. 4c and analyze the reaction rate by microkinetic modelling and make comparison between ammonia and nitrite productions.

7. The recent experiments (10.1002/anie.202303327) found the electrochemical performance of nitrate conversion can be enhanced by tandem process. If the present results are contributed from the second electroreduction of as-produced nitrite?

8. The reviewer did not see any descriptions regarding the solvation effects of adsorbates. It seems all adsorption energies are simply calculated by models in vacuum?

All the concerns need to be solved before consideration of publication!

Reviewer #2 (Remarks to the Author):

In the manuscript of “Defective Cu Enabled Triple Synergistic Modulation Endows Superior Electrochemical Ammonia Production at Wide-Range Nitrate Concentrations”, the defect-rich copper nanowires exhibited high current density and Faraday efficiency across a wide range of nitrate concentrations, reducing nitrate levels to meet WHO drinking water standards, and leading in performance. Through the authors’ detailed exploration on the triple synergistic modulation induced by copper defects, employing both synchrotron in-situ infrared and DFT calculations, authors have successfully unveiled the mechanisms that contribute to the superior electrochemical ammonia production. Finally, through a two-electrode system, authors achieved impressive nitrate degradation and ammonia production capabilities at low cell voltages, further expanding its application. Overall, the authors’ work provides inspiration for solving the degradation and ammonia production of nitrate pollutants under real conditions. I recommend the acceptance of this manuscript after minor revisions. The issues that need to be addressed are listed as follows:

1. The field of electrocatalytic nitrate reduction to ammonia is very promising and active. In this research, have the authors considered the high energy consumption issue associated with the eight-electron transfer process typically required for nitrate ion reduction? Additionally, compared to the established and widely applied Haber-Bosch process, does the method have sufficient cost-effectiveness to offset this high electrical energy cost? If so, could authors provide supportive data or models to prove the cost-benefit advantages of the electrocatalytic nitrate reduction method? These are key factors in evaluating the feasibility of an electrocatalytic nitrate reduction method, and I look forward to authors’ detailed response.
2. To demonstrate the performance enhancement of copper in a wide nitrate concentration range due to defect introduction, I hope authors could set up further comparisons, testing pure copper without defects across a broad range of nitrate concentrations. Moreover, the authors should also supplement the performance of other comparative samples at different potentials.
3. More experimental data to support catalyst durability would be appreciated. After continuous cycle tests, is the catalyst stable in the nitrate reduction process? Are there any changes in its phase and morphology? What are the reasons for catalyst performance degradation? More characterization and comparisons for the catalyst before and after measurements are needed.
4. The authors have performed some in-situ characterizations, such as synchrotron in-situ infrared, in situ Raman spectroscopy, which are very valuable for research as they can intuitively display the inherent characteristics of substances under specific conditions. However, presenting these data in the form of color mapping (Fig. 1 j and Fig. 3 a), without color labels for specific numerical expression, may lead to loss or misinterpretation of information.
5. From a reviewer's perspective, it is crucial to reflect the use of an electrochemical model in DFT calculations, rather than relying solely on thermodynamics calculations. Specifically, it's important to account for the significant influence of the electrode potential on the rate-determining energy barrier for NO₃⁻ reduction reaction (NO₃RR). This reaction involves the adsorption of NO₃⁻ and proton - electron transfer, both of which exhibit contrasting responses to potential. With this in mind, the authors should indicate the voltage corresponding to the experimentally optimal performance, and present a free energy diagram of NO₃RR from an electrocatalytic viewpoint. This would greatly aid readers in gaining a comprehensive understanding of the

process, and in evaluating the effectiveness and applicability of the model. As a reviewer, I perceive this to be a critical component of understanding the research results, and also a reflection of the depth and quality of the author's investigation. Furthermore, the energies calculated for all intermediates should be included in the supplementary information tables.

6. The authors integrated the nitrate reduction reaction and glycerol oxidation reaction in wastewater through a two-electrode system and provided detailed analysis of nitrate reduction at the cathode end. However, analysis on the yield and Faraday efficiency of glycerol oxidation at the anode end is missing. Furthermore, during the gas stripping process, the authors used air to blow out the ammonia product instead of using an inert gas. Could this have an effect on the collection of the ammonia product?

Reviewer #3 (Remarks to the Author):

In this study, a defect-rich Cu nanowire array electrode (V-Cu NAE) was fabricated, which exhibited superior performance in the electrochemical nitrate reduction reaction (NO₃RR) across a wide range of NO₃⁻ concentrations (1-100 mM) with high FEs, NH₃ yields, and NO₃⁻ conversions. In-situ characterization and DFT calculations indicate that the triple synergistic modulation on the defective Cu sites simultaneously enhances nitrate adsorption, promotes water dissociation, and suppresses hydrogen evolution, resulting in this high performance. Additionally, a two-electrode system was established, which integrated the NO₃RR in industrial wastewater with the glycerol oxidation reaction, further demonstrating its practicality. The manuscript is well organized, and the results are credible. However, some data still deserves further analysis. To meet the high quality of Nat. Commun., major revisions are required. Here are some comments for authors:

1. The interpretation of the operando SR-FTIR spectroscopy (Fig. 3) could benefit from additional evidence to support it. It is particularly worth discussing that the typical absorption peak of water is located around 3400 cm⁻¹. Therefore, it is important to consider the possibility of water absorptions in this region and distinguish them from the free ammonia signals. Conducting the same test in a solution without NO₃⁻ reactants can indeed help to distinguish the contribution of NH₃ in the observed spectra.
2. Although images may undergo compression when converted into documents, some images remain excessively blurry. For example, the yellow and red arrows in Fig. 1e are not prominent. Moreover, the small white circle mentioned on page 7, line 105, may lead to misunderstanding. It is suggested to replace it with a different color to enhance contrast.
3. To analyze the impact of Cu vacancies on the surface properties of the samples, it is necessary to supplement the XPS data of Cu NWs that without vacancies.
4. The electrochemical testing conditions, such as the applied potential, are not clearly stated after Fig. 2b. Detailed information needs to be provided to enhance the rigor of the study.
5. The WHO regulations for drinking water have been mentioned repeatedly.
6. The results of SEM, XRD or other related characterizations of the samples after stability test should also be provided. Evidence that can reflect the changes in Cu vacancies should be carefully analyzed.

Responds to Reviewers

Reviewer #1: Zhang et al. reported a combined experimental and theoretical work to study the electrochemical ammonia synthesis via nitrate electroreduction. The defect-rich Cu nanowire array electrode (V-Cu NAE) was generated by in-situ electrochemical reduction. Operando Synchrotron radiation-FTIR and density functional theory (DFT) calculations were used to explain these issues. It claims there are synergy over V-Cu NAE to simultaneously enhance NO₃RR activity, suppressing HER, and promoted water dissociation. Overall, there are two important achievements including the activity of ammonia production at varying nitrate concentration and the extraction of ammonia products from industrial wastewater. However, there are plenty of fundamental issues unclear.

Author response: Firstly, we really appreciate the Reviewer for the thorough review and valuable suggestions on our manuscript, particularly in the domain of theoretical calculations. The reviewer's profound insights have been of immense benefit to our research. According to the overall suggestions of the reviewers, we adjusted the model for the DFT calculation as well as the details of the calculation. The details are as follows:

- a. According to the thermodynamic cycle, we have detailed how the HNO₃ energy is used as a reference for nitrate adsorption energy, and nitrate adsorption is still potential dependent.
- b. Based on the original model, four explicit water molecules were added to account for the role of hydrogen bonds, while the implicit solvation model was adopted.
- c. According to the SCE reduction potential (-1.3V) corresponding to the best performance, the influence of the electrode potential on the selectivity of the reaction path, the speed step and the energy barrier is discussed after converting it to the RHE scale.
- d. In order to more accurately reflect the actual reaction potential conditions, the constant potential model of the double reference method with implicit solvation is considered. This is equivalent to introduce an external electric field.

- e. Full consideration has been given to the possible intermediates involved, and the desorption process of nitrite and ammonia has been taken into account.
- f. The slow-growth method was used to calculate the transition states of non-electrochemical and electrochemical elementary steps, considering the potential dependent reaction energy and activation energy, and MKMCXX software was used to carry out microscopic dynamics modeling to investigate the influence of Cu vacancy defects on the yield of nitrite and ammonia at low voltage.

Before starting to reply to your suggestions, the necessary calculation method is provided below for your convenience.

Method R1. Potential dependent reaction energy

Nørskov's computational hydrogen electrode (CHE) method (Phys. Rev. Lett. 2007, 99, 126101; J. Phys. Chem. B 2004, 108, 17886–17892) is utilized to describe the chemical potential of protons and electrons. Specifically, using the linear free energy dependency on electronic energy under this potential, the free energy dependency of the proton-electron pair on the electrode potential is determined, that is, the electronic energy is offset by $-eU$.

$$\mu(H^+) + \mu(e^-) = \frac{1}{2}\mu(H_2) - eU \quad (M1)$$

Here, e represents the elementary positive charge ($1.602176634 \times 10^{-19}$ C), and U is the electrode potential relative to Reversible Hydrogen Electrode (RHE). For the protonation process i , its free energy barrier ΔG_i will undergo the following changes:

$$\Delta G_i(U) = \Delta G_i(U = 0 \text{ vs. RHE}) + eU \quad (M2)$$

Where, U is the electrode potential relative to the RHE. Under a more negative reduction voltage, the protonation process becomes increasingly favorable.

For the adsorption process of the nitrate ion and the desorption process of the nitrite ion, their free energy barriers will undergo the following changes:

$$\Delta G(NO_3) = \Delta G(NO_3) - eU \quad (M3)$$

$$\Delta G(NO_2) = \Delta G(NO_2) - eU \quad (M4)$$

Under a more negative reduction voltage, the adsorption process of the nitrate ion

becomes increasingly difficult, while the desorption process of the nitrite ion becomes increasingly favorable. At the same time, the protonation process becomes relatively easier.

Method R2. Constant potential simulation

To more precisely account for actual reaction conditions, such as pH and electrode potential, we adopted the double-referencing method developed by Duan et al. (ACS Catal. 2019, 9, 5567–5573) to study the energetics of NO₃RR. This approach elucidates the intrinsic catalytic active sites and pH-dependent activity of NO₃RR under actual reaction conditions. The implicit solvation model is implemented through VASPsol (J. Chem. Phys. 140, 084106 (2014); J. Chem. Phys. 151, 234101 (2019)). A relative dielectric constant of 80 was set to simulate the water electrolyte. VASPsol assigns an effective surface tension parameter value of 0, thereby neglecting the contribution of cavitation energy. The compensating charge was simulated by a linearized Poisson-Boltzmann model with a Debye length of 3.0 Å. To clarify the reaction mechanism under different electrode potentials, we adjusted varying amounts of excess charge (q), varying from -2.0 e to +2.0 e in increments of +0.5 e .

The free energy under the potential-dependent grand canonical ensemble is as follows:

$$E_{free}(U) = E_{DFT} + \int_0^q |\overline{V}_{tot}| dQ + qW_f/e \quad (\text{M5})$$

Where E_{free} represents the free energy under the grand canonical ensemble, E_{DFT} is the energy derived from DFT calculations, $|\overline{V}_{tot}|$ is the average electrostatic potential of the model, W_f is the work function of the charged system, and q is the number of electrons doped into the system.

The electrode potential relative to the standard hydrogen electrode (SHE) can be determined from the system's work function W_f .

$$U_{SHE} = W_f/e - 4.6 \text{ V} \quad (\text{M6})$$

Where 4.6 V is the absolute electrode potential reference of the standard hydrogen electrode (SHE) in VASPsol.

A quadratic function relationship exists between E_{free} and U_{SHE}

$$E_{free}(U_{SHE}) = -\frac{1}{2}C(U_{SHE} - U_{PZC})^2 + E_{PZC} = aU_{SHE}^2 + bU_{SHE} + c \quad (M7)$$

Where C represents the capacitance of the system. U_{PZC} and E_{PZC} are the electrode potential and free energy, respectively, relative to SHE in a zero-charge system. The vertex form of the quadratic function relationship can be simplified to the standard form of a quadratic function, where a , b , and c are parameters that need to be fitted. After obtaining E_{free} under actual reaction conditions, E_{DFT} in equation S13 can be replaced with E_{free} to simulate the free energy under constant potential. The parameters fitted for all systems involved in Cu and V-Cu are presented in Tables R1 and R2.

Method R3. Potential dependent activation energy barrier

The potential dependent activation energy can be treated following the method of Akhade et al. (Catal. Today 2017, 288, 63–73). For non-electrochemical surface protonation:

The activation energy (E_a) can be obtained by DFT calculations of the reactant's minimum-energy geometry and the corresponding hydrogenation transition state. For the corresponding electrochemical step, at equilibrium $H^+ + e^- + * \rightleftharpoons H^*$ conditions, the activation energy $E_a(U^0)$ for $*A + H^+ + e^- \rightleftharpoons AH^*$ equals E_a . U^0 is the equilibrium potential at which the analogous nonelectrochemical state, $\mu(*H)$, is in equilibrium with its equivalent electrochemical state, $\mu(H^+(aq) + e^-)$. (Catal. Today 2017, 288, 63–73) Here, U^0 equals the hydrogen adsorption free energy (ΔG_H) for a given surface at 0 V vs RHE. The free energy change for the electrochemical surface hydrogenation can then be computed by $U^0 = [G(*A+*H) - G(*A) - 0.5*G(H_2)]/e$. We assume that the forward activation energy at an electrode potential U follows the Butler–Volmer formalism

$$E_a(U) = E_a(U^0) + \alpha ne(U - U^0) \quad (M9)$$

where n is the number of electrons transferred. In the current work, α was set to 0.5 for all elementary steps, which has been shown to often be a reasonable approximation. (J.

Method R4 Microkinetic Simulations

The intermediate reactions considered are:

For the surface reactions, the rate constants for the forward and backward elementary reactions are determined by the Arrhenius equation:

$$k_i^+ = A e^{-\frac{E_a}{k_b T}} \quad (\text{M10})$$

$$k_i^- = A e^{-\frac{E_b}{k_b T}} \quad (\text{M11})$$

where k_i is the rate constant in s^{-1} for elementary step i , A is the pre-exponential factor, E_a is the activation energy, T is the temperature, and k_b is Boltzmann's constant. A was approximated as 10^{13} s^{-1} for all the elementary surface reactions (PCCP 2013, 15, 17038-17063). For non-activated molecular adsorption, the rate of adsorption was determined by the rate of surface impingement of gas-phase molecules. Based on the Hertz-Knudsen equation (Nanoscale 2013, 5, 9732-9738), the molecular adsorption rate constant of species i was computed as:

$$k_{ads} = \frac{pA'}{\sqrt{2\pi m k_b T}} S \quad (\text{M12})$$

where p is the partial pressure of the adsorbate in the gas phase, A' the surface area of the adsorption site, m the mass of the adsorbate, and S the sticking coefficient, which we assume takes a value of unity for all adsorbates. k_{11}^+ , k_{18}^- and k_{19}^- should be calculated according to Eq. M12.

For molecular desorption, we assumed there are three rotational degrees of freedom

and two translational degrees of freedom in the activated state. Accordingly, the rate constant of desorption for adsorbate i was calculated as:

$$k_{des} = \frac{k_b T^3}{h^3} \frac{A' (2\pi k_b)}{\sigma \theta_{rot}} e^{-\frac{E_{des}}{k_b T}} \quad (M13)$$

where E_{des} is the desorption energy, h is Planck's constant, and σ and θ_{rot} are the symmetry number and the characteristic temperature for rotation, respectively. k_1^- , k_{18}^+ and k_{19}^+ should be calculated according to Eq. M13.

The nitrate reduction rate was calculated by the MKMCXX microkinetic modeling software suite for heterogeneous catalysis (Angew. Chem. 2014, 126, 12960-12964; ACS Catal. 2015, 5, 5453-5467). In our simulations, the molar ratio of NO_3^- and H^+ in the solution was 1:1 at a reaction temperature of 300 K. For each of the M components in the kinetic network, a single differential equation for each elementary reaction step was written in the form of:

$$r_i = k_i \prod_{w=1}^M c_w^{v_w^i} \quad (M14)$$

where k_i is the rate constant and c_w and v_w^i are the concentration and stoichiometric coefficient of species w in elementary reaction step i . Specifically, all the involved constraints are as follows:

$$r_1 = k_1^+ P_{NO_3^-} \theta_* - k_1^- \theta_{*NO_3} \quad (M15)$$

$$r_3 = k_3^+ \theta_{*NO} P_{H^+} - k_3^- \theta_{*NOH} \quad (M16)$$

$$r_4 = k_4^+ \theta_{*NO} P_{H^+} - k_4^- \theta_{*NHO} \quad (M17)$$

$$r_5 = k_5^+ \theta_{*NOH} P_{H^+} - k_5^- \theta_{*NHOH} \quad (M18)$$

$$r_6 = k_6^+ \theta_{*NHO} P_{H^+} - k_6^- \theta_{*NHOH} \quad (M19)$$

$$r_7 = k_7^+ \theta_{*NHO} P_{H^+} - k_7^- \theta_{*NH_2O} \quad (M20)$$

$$r_8 = k_8^+ \theta_{*NHOH} P_{H^+} - k_8^- \theta_{*NH} P_{H_2O} \quad (M21)$$

$$r_9 = k_9^+ \theta_{*NHOH} P_{H^+} - k_9^- \theta_{*NH_2OH} \quad (M22)$$

$$r_{10} = k_{10}^+ \theta_{*NH_2O} P_{H^+} - k_{10}^- \theta_{*NH_2OH} \quad (M23)$$

$$r_{11} = k_{11}^+ \theta_{*NOH} P_{H^+} - k_{11}^- \theta_{*N} P_{H_2O} \quad (M24)$$

$$r_{12} = k_{12}^+ \theta_{*N} P_{H^+} - k_{12}^- \theta_{*NH} \quad (\text{M25})$$

$$r_{13} = k_{13}^+ \theta_{*NH} P_{H^+} - k_{13}^- \theta_{*NH_2} \quad (\text{M26})$$

$$r_{14} = k_{14}^+ \theta_{*NH_2OH} P_{H^+} - k_{14}^- \theta_{*NH_2} P_{H_2O} \quad (\text{M27})$$

$$r_{15} = k_{15}^+ \theta_{*NH_2} P_{H^+} - k_{15}^- \theta_{*NH_3} \quad (\text{M28})$$

$$r_{16} = k_{16}^+ \theta_{*O} P_{H^+} - k_{16}^- \theta_{*OH} \quad (\text{M29})$$

$$r_{17} = k_{17}^+ \theta_{*OH} P_{H^+} - k_{17}^- \theta_{*H_2O} \quad (\text{M30})$$

$$r_{18} = k_{18}^+ \theta_{*NO_2} - k_{18}^- \theta_{*NO_2^-} \quad (\text{M31})$$

$$r_{19} = k_{19}^+ \theta_{*NH_3} - k_{19}^- \theta_{*NH_3} \quad (\text{M32})$$

$$\theta_{*} + \theta_{*NO_3} + \theta_{*NO_2} + \theta_{*NO} + \theta_{*NHO} + \theta_{*NOH} + \theta_{*NHOH} + \theta_{*NH_2O} + \theta_{*N} + \theta_{*NH} + \theta_{*NH_2} + \theta_{*NH_3} + \theta_{*O} + \theta_{*OH} = 1 \quad (\text{M33})$$

Steady-state coverages were computed by integrating the ordinary differential equations in time until changes in the surface coverages were small ($< 10^{-12}$). Rates of the individual elementary steps were obtained based on the computed steady-state surface coverages.

Method R5 Slow-growth method

Both the AIMD and Slow-growth approach simulations were sampled within the canonical (NVT) ensemble by Nosé-Hoover thermostats with a time step of 1.0 fs at a finite temperature of 300 K.

we mainly adopted the Slow-growth method (J. Phys. Chem. B, 101 (40) (1997) pp. 7877-7880; J. Phys. Chem. B, 109 (14) (2005), pp. 6902-6915) to obtain the free energy barrier. In the Slow-growth method, a geometric parameter λ , i.e., the collective variable (CV), is constrained during the dynamics and varied from the value characteristic of the initial state (λ_0) to that of the final state (λ_1) with a velocity of transformation v . The Helmholtz free energy difference ΔF can be computed as Equation (1) by collecting the derivative of the potential energy with respect to lambda during the simulation, and by integrating over lambda afterwards.

$$\Delta F = \int_{\lambda_0}^{\lambda_1} \left(\frac{\partial V(q)}{\partial \lambda} \right) \cdot v dt \quad (\text{M34})$$

In the limit of infinitesimally small v , ΔF corresponds to the free energy difference ΔG between the final and initial state. The parameters of the method for different elementary reactions are presented in Tables R3 and R4. The transition states determined for all elementary reactions of the NO_3RR path for Cu and V-Cu are presented in Supplementary Fig. 20 and Supplementary Fig. 21.

1. The authors chose to refer to the HNO_3 molecule in studying the first step of nitrate conversion. As the nitrate is negatively charged, it will become more and more difficult at reducing potentials. In other words, the adsorption energies of nitrate will be potential dependent and become more and more positive (at more reducing potentials). Instead, the adsorption of HNO_3 should be more favorable and independent on electrode potentials. The authors should carefully check the literature and clarify how the nitrate was converted at the first step.

Author response: We thank the reviewer for the professional suggestions, which is very important for us to modify our manuscript. We totally agree with your point that the nitrate anion carries a negative charge, and therefore, the adsorption of nitrate will be suppressed at more negative reduction potentials. However, using the energy of the HNO_3 molecule as a reference is consistent with the fact that nitrate becomes increasingly difficult to adsorb under reduction potentials. This is because the adsorption energy of nitrate is not directly referenced to the energy of HNO_3 , but is indirectly obtained through a thermodynamic cycle, which merely involves the energy of HNO_3 (Journal of Catalysis, 2021, 402,1-9; Adv. Funct. Mater., 2021, 31, 2008533).

Specifically, for DFT calculation, obtaining the energy of the nitrate is challenging because DFT cannot accurately obtain the energy of charged molecules. In this context, the adsorption energy of the nitrate (Scheme 1) is generally obtained using thermodynamic data from the CRC Handbook combined with the accurate energy that can be directly calculated based on DFT. This is how we handled it for the related calculation results described in our original manuscript, while we are very sorry that we

did not detail this aspect in the supporting information or main text. These details has been supplemented in the computational details section of the supporting information.

The specific derivation process is provided as follows:

According to Hess's law, S4 represents the adsorption process of the nitrate ion, and the Gibbs free energy of S4 can be derived from S1, S2, and S3, namely:

$$\Delta G_{S4} = \Delta G(NO_3) = \Delta G_{S1} + \Delta G_{S2} + \Delta G_{S3} \quad (S5)$$

Where ΔG_{S8} and ΔG_{S9} can be obtained by referring to the CRC Handbook of Chemistry and Physics, with values of 0.317 eV and 0.075 eV, respectively. ΔG_{S3} can be derived from DFT calculations.

$$\Delta G_{S3} = \Delta G_{ads}(*NO_3) = G(*NO_3) + \mu(H^+) + \mu(e^-) - G(HNO_3(g)) - G(*) \quad (S6)$$

Where $G(*NO_3)$, $G(HNO_3(g))$ and $G(*)$ are the free energies of the system after NO_3 adsorption on the slab, the free energy of the gaseous HNO_3 molecule, and the free energy of the slab, respectively. $\mu(H^+)$ and $\mu(e^-)$ represent the chemical potentials of H^+ and e^- , respectively. The calculation of ΔG_{S3} involves the chemical potentials of protons and electrons. Therefore, even when the energy of HNO_3 is used as a reference for calculating nitrate adsorption, this process remains potential-dependent.

Additionally, to consider the desorption of nitrite, the energy of nitrite is also required. Similarly, the energy of nitrite is derived using thermodynamic data from the CRC Handbook combined with DFT for accurate energy calculations.

According to Hess's law, the Gibbs free energy of S7 can be derived from S8, S9,

and S10, namely:

$$\Delta G_{S7} = \Delta G(NO_2) = \Delta G_{S8} + \Delta G_{S9} + \Delta G_{S10} \quad (S11)$$

Where ΔG_{S1} and ΔG_{S2} can be obtained by referring to the CRC Handbook of Chemistry and Physics, with values of -0.198 eV and 0.069 eV, respectively. ΔG_{S10} can be derived from DFT calculations.

$$\Delta G_{S10} = \Delta G_{ads}(*NO_2) = G(*NO_2) + \mu(H^+) + \mu(e^-) - G(HNO_2(g)) - G(*) \quad (S12)$$

Where $G(*NO_2)$, $G(HNO_2(g))$ and $G(*)$ are the free energies of the system after NO_3 adsorption on the slab, the free energy of the gaseous HNO_3 molecule, and the free energy of the slab, respectively. $\mu(H^+)$ and $\mu(e^-)$ represent the chemical potentials of H^+ and e^- , respectively.

The free energy consists of three terms, as shown in Equation S13, which are the DFT computed energy (E_{DFT}), zero-point energy (ZPE), and the contribution from entropy change (TS):

$$G = E_{DFT} + ZPE - TS \quad (S13)$$

In this case, the temperature T is taken as 298.15K.

Scheme 1. The thermodynamic cycle used to calculate the adsorption Gibbs free energy of NO_3^- in the gas phase. The thermodynamic values indicated are obtained from the CRC handbook of chemistry and physics (CRC Handbook of Chemistry and Physics, CRC Press, 90th Ed. 2010).

Scheme 2. The thermodynamic cycle used to calculate the adsorption Gibbs free energy of NO_2^- in the gas phase. The thermodynamic values indicated are obtained from the CRC handbook of chemistry and physics (CRC Handbook of Chemistry and Physics, CRC Press, 90th Ed. 2010).

2. It is obviously incorrect to describe the hydrogen evolution by water dissociation on the surface because it is a chemical process without electron-proton transfer between reaction plane and electrode. At the reducing potentials, the electrochemical protonation by either hydronium or water should be faster. Based on the proposed mechanism shown in Fig. 4b, the HER performance should be hard to be affected by electrode potentials, while it is against lots of experiments. BTW, what is the pH value at the electrochemical interface? Please also convert the SCE to RHE scale and be more friendly for readers.

Author response: Thank you very much for carefully reviewing our research and raising important questions about the hydrogen evolution reaction mechanism. Your view, namely that describing hydrogen evolution through surface hydrolysis is apparently incorrect, has drawn our attention. At the same time, we indeed realize that based on the mechanism shown in Fig. 4b, the impact of electrode potential on HER performance might not align with many experimental results.

To simulate the conditions that more closely resemble actual conditions, the CHE model and constant potential simulation (Methods R1 and R2) were used to incorporate

the effect of electrode potential on adsorption energy. Under the grand canonical ensemble, the free energy E_{free} at different electrode potentials is calculated (Supplementary Fig. 19). All system energies are affected by electrode potential. It is noteworthy that the energy of two systems involving the water cleavage energy, namely $^*\text{H}_2\text{O}$ and $^*\text{OH} + ^*\text{H}$, is also affected by the variation in electrode potential. This indicates that the reaction energy of non-electrochemical processes is still affected by the electrode potential and is not entirely potential-independent, which has also been reported in previous literature (J. Phys. Chem. C 2021, 125, 15243). Free energy diagrams for water cleavage and HER will be shown separately (Fig. 4). Water cleavage is still treated as a non-electrochemical process, with the transition state energy barrier obtained using the slow-growth method (Method R5). At the equilibrium condition of $\text{H}^+ + \text{e}^- + ^* \rightleftharpoons \text{H}$, treating it as $^*\text{H}_2\text{O} \rightleftharpoons ^*\text{OH} + (\text{H}^+ + \text{e}^-)$ is feasible (Method R3). HER is considered as an electrochemical process, and its free energy is affected by electrode potential (Method R1 and R2). At an applied voltage of $U_{\text{RHE}} = -0.289$ V, the water cleavage activation energy of V-Cu is only 0.38 eV, which is lower than the water cleavage activation energy of Cu (0.72 eV), indicating that V-Cu is more conducive to water cleavage and provides more protons (Fig. 4b). Additionally, the HER results show that V-Cu has a more negative adsorption energy for $^*\text{H}$, indicating its stronger ability to suppress HER.

Our experimental results show that the sample with Cu defects performs best under an applied voltage of $U = -1.3$ V vs. SCE. We assume that the pH is maintained at around 13 throughout the reaction process, corresponding to U_{SHE} of -1.059 V, and using U_{SHE} makes it easier to get E_{free} (Method R2). After converting the applied voltage from SCE to RHE, the U_{RHE} here is -0.289 V, and U_{RHE} will be more convenient for obtaining the potential-dependent reaction energy (Method R1).

Supplementary Figure 19. Constant potential simulation of water splitting and HER.

The $E_{\text{free}}(U_{\text{SHE}})$ of $\text{*H}_2\text{O}$, $\text{*OH} + \text{*H}$ systems for (a) Cu and (b) V-Cu.

Fig. 4. DFT Calculations. (b) Depicted water splitting reaction pathways for both V-Cu and Cu when $U = -0.289$ V vs RHE. (c) Showcases the hydrogen evolution reaction pathways for V-Cu and Cu at $U = -0.289$ V vs RHE.

Based on previous studies (Nat. Commun. 2023, 14, 4209), the pH values at the electrochemical interface were determined using the Rotating Ring-Disk Electrode (RRDE) technique. The potential of the Pt ring electrode (RE) is sensitive to pH, allowing for monitoring of pH variations on the surface of the disk electrode (DE). A three-electrode cell was assembled using the RRDE (AFE6R2 with a collection efficiency of $N_D=0.38$), a graphite rod, and a saturated calomel electrode (SCE) serving as the working, counter, and reference electrodes, respectively. Subsequently, the pH-

dependent open circuit potential (E_{ocp}) of the Pt RE was measured in an H_2 -saturated K_2SO_4 solution (as seen in Supplementary Fig. 19). The OCP of the Pt electrode signifies the equilibrium potential for the reaction $2H^+ + 2e^- \rightarrow H_2$, and this potential varies with pH as dictated by the Nernst equation:

$$E(\text{V vs. SHE}) = \frac{-2.303RT}{F} \text{pH}$$

Wherein, the fugacity of H_2 is assumed to be unity. R , T , and F denote the gas constant, absolute temperature, and Faraday constant, respectively.

Fig. R1. (a) Open circuit potential of the Pt ring electrode as a function of time under various pH conditions. (b) pH dependency of the Pt ring electrode's open circuit potential after fitting.

For pH measurements at the electrochemical interface, the catalyst under investigation was loaded onto the disk electrode. Measurements of the catalyst surface pH were conducted in a K_2SO_4 solution with 200 ppm NO_3^- -N, with the working electrode rotating at a speed of 1600 rpm. Electrocatalytic reactions were carried out on the disk electrode to achieve a steady-state current response, while the OCP was simultaneously measured on the Pt ring electrode. The pH value on the catalyst-loaded DE can be inferred from the pH value of the Pt RE, as given by the relation:

$$c_{r,OH^-} = N_D \cdot c_{d,OH^-}$$

Upon conducting electrocatalytic nitrate reduction tests at the optimal potential, we found that the pH at the electrochemical interface was approximately 13.6. This increase in pH may be attributed to reactions taking place on the surface of the catalyst, leading to an increase in the local alkalinity.

In addition, as reviewer pointed out, converting from SCE to RHE scale indeed makes the content more reader-friendly. According to the reviewer's suggestions, we have adjusted the data according to the equation: $E_{RHE} = E_{SCE} + 1.0$.

We thank the reviewer again for the professional questions that are quite significant for us to revise the manuscript.

3. Why the proton has to be transferred to O*, instead of via NHO*, at the first protonation? Why NH₂OH* was considered in the mechanism study? Why the NH₃* was not desorbed to complete a catalytic cycle? The whole process is potential dependent of ammonia production, what is the potentials at Fig. 4d. How the chemical potential of proton and electron were calculated for these equations listed in S1-S8?

Author response: We thank the reviewer for the insightful questions. The nitrate reduction reaction process involves multiple proton-electron transfers, adding complexity to the reaction pathway, especially the question of whether the proton should be transferred to N or O. The pathway in Fig. 4c is consistent with previous work by Hu et al. (ACS Catal. 2021, 11, 14417–14427), where their findings suggest that it is easier for NO₃ and NO₂ to directly deoxygenate than protonate, but the opposite is true for NO. As you pointed out, during the reduction of NO as the first protonation process, the proton might transfer to either O or N. This crucial judgment was not detailed in the work by Hu et al. We apologize for our oversight. In the revised manuscript, we have considered all possible intermediates involved in NO₃RR as thoroughly as possible (Supplementary Fig. 18), including the adsorption and desorption of NO₃, NO₂, and NH₃, and the protonation process starting from NO. We will further consider the dominant intermediates in the reaction process from both kinetic and thermodynamic perspectives.

In answer to question 2, we provided the electrode potential used in the simulations. "Our experimental results show that under an applied voltage of $U = -1.3$ V vs. SCE, the sample with Cu defects performs best. We assume that throughout the reaction process, the pH is maintained around 13, corresponding to U_{SHE} of -1.059 V; using U_{SHE} makes obtaining E_{free} more convenient (Method R2). After converting the applied voltage from

SCE to RHE, the U_{RHE} here is -0.289 V, making U_{RHE} more suitable for obtaining potential-dependent reaction energy (Method R1)." (from question 2)

According to the path shown in Supplementary Fig. 18, we recalculated the reaction free energy for NO_3RR . For all intermediates involved in NO_3RR , the relationship between their E_{free} and U_{SHE} was calculated (Method R2), with the results shown in Supplementary Fig. 19. According to the results in Fig. 4, under conditions of $U = -0.289$ V vs RHE and based on kinetic favorability, we determined the final reaction pathway to be $\text{NO}_3^-(l) \rightarrow * \text{NO}_3 \rightarrow * \text{NO}_2 \rightarrow * \text{NO} \rightarrow * \text{NHO} \rightarrow * \text{NH}_2\text{O} \rightarrow * \text{NH}_2\text{OH} \rightarrow * \text{NH}_2 \rightarrow * \text{NH}_3 \rightarrow \text{NH}_3(g)$. From both kinetic and thermodynamic perspectives, the proton in the first protonation step should be transferred to N rather than O. Additionally, $* \text{NH}_2\text{OH}$ does appear in the final reaction pathway, and the desorption of NH_3 has been reconsidered to ensure a complete catalytic process. Regarding the analysis of Fig. 4d and Fig. 4e, we will discuss it in more detail in question 6, combined with the results of microscopic kinetic simulations.

Supplementary Figure 18. The reaction mechanism of NO_3RR and the intermediates involved. The desorption of nitrite and ammonia and protonation starting from NO are considered.

Supplementary Figure 19. Constant potential simulation of NO_3RR . The $E_{\text{free}}(U_{\text{SHE}})$ of slab, $* \text{O}$, $* \text{N}$, $* \text{OH}$, $* \text{NH}$, $* \text{NO}$, $* \text{NH}_2$, $* \text{NOH}$, $* \text{NHO}$, $* \text{NO}_2$, $* \text{NH}_3$, $* \text{NHOH}$, $* \text{NH}_2\text{O}$, $* \text{NH}_2\text{OH}$ and $* \text{NO}_3$ systems for (a) Cu and (b) V-Cu.

Fig. 4. DFT Calculations. (d) The NO₃RR pathways for Cu are displayed, with the optimal pathway distinctly highlighted in blue at $U = -0.289$ V vs RHE. (e) The NO₃RR pathways for V-Cu are demonstrated with the optimal pathway prominently marked in red at $U = -0.289$ V vs RHE.

4. Besides the chemical potential of electron and protons, the adsorption energies of species can also be affected by electric fields. How much is the electric field effects on the adsorption energies shown in Fig 4.

Author response: We sincerely appreciate your insight and the opportunity to discuss this intriguing phenomenon. From a computational perspective, there are two main methods that can be used to take an external electric field into account in a VASP-implemented DFT calculation. One method, the so-called 'double reference method' (Angew. Chem. Int. Ed. 2006, 45, 402-406; Phys. Rev. B 2006, 73, 165402), was developed by the Neurock group. This method generates an external electric field by adding (subtracting) explicit charges to (from) electrode-electrolyte interface (RSC 2014, pp. 116-156). From this method, one can calculate the electrode potential from

the energy differences between the calculated work function of the charged system and the absolute standard hydrogen electrode (SHE) from experimental work (J. Electroanal. Chem. Interfacial Electrochem. 1984, 179, 71-82). The other method, the 'Neugebauer and Scheffler (NS) method' (Phys. Rev. B 1992, 46, 16067-16080), includes a dipole sheet that is negatively (positively) charged at one end of the sheet and positively (negatively) charged at the other end of the sheet. Here, we adopt the first approach, the dual reference method under the implicit solvent model developed by Duan et al (ACS Catal. 2019, 9, 5567–5573). At the same time, the explicit water molecules in the model were removed to better understand the effect of the electric field on the adsorption energy.

We computed the energy profiles of Cu and V-Cu slabs as well as *NO₃ and *NO₂ as a function of electrode potential, following the methodology in Method R2 (as seen in Fig. R3a and Fig. R3b). The adsorption energy for NO₃ and NO₂ on the catalyst is derived using the following formula:

$$E_{ads}(*NO_x, x = 2,3) = E_{free}(*NO_x, U_{SHE}) - E_{free}(slab, U_{SHE}) - E_{DFT}(NO_x(g)) \quad (S14)$$

Here, $E_{ads}(*NO_x, x = 2,3)$ represents the adsorption energy of *NO_x, while $E_{free}(*NO_x, U_{SHE})$ and $E_{free}(slab, U_{SHE})$ represent the energies of *NO_x and the slab under the grand canonical ensemble, respectively. $E_{DFT}(NO_x(g))$ denotes the DFT energy of the NO_x in the gaseous state. The use of $E_{DFT}(NO_x(g))$ is to simplify the computation of the adsorption energy. As this value remains potential-independent, it does not influence the relationship between the electrode potential and adsorption energy.

The relationship between the adsorption energy and the electrode potential is depicted in Fig. R3c. Notably, as the electrode potential decreases, the adsorption strength of both Cu and V-Cu towards NO₃ and NO₂ progressively intensifies. This trend is in line with numerous studies using constant potential models (ACS Catal. 2019, 9, 5567–5573; J. Phys. Chem. Lett., 2021, 12(51): 12230-12234). Within the electrode potential range of -1.5 V to 1.5 V, the results indicate that under reductive potentials (more negative electrode potentials), the enhancement in adsorption energy computed

under constant potential will offset some of the positive corrections to adsorption energy introduced by the CHE model. For both Cu and V-Cu, the change rate of the adsorption energy of NO_3 with respect to electrode potential is 0.445 eV/V . In contrast, the change rates for the adsorption energy of NO_2 are 0.294 eV/V for Cu and 0.314 eV/V for V-Cu. From the results, it is evident that the adsorption energy of NO_3 is more sensitive to changes in electrode potential compared to that of NO_2 . Interestingly, the impact of the electrode potential on the adsorption energy seems to be largely substrate-independent, being determined mainly by the type of intermediate species.

Fig. R2. The surface electrode potential is modulated by adding and removing electrons to the system. When additional electrons are added, the electrode potential on the catalyst surface decreases. When the extra electrons are removed, the electrode potential on the catalyst surface rises.

Fig. R3. Effect of electric field on adsorption energy. The $E_{\text{free}}(U_{\text{SHE}})$ of slab, $^*\text{NO}_2$ and $^*\text{NO}_3$ systems for (a) Cu and (b) V-Cu. (c) The $E_{\text{ads}}(U_{\text{SHE}})$ of $^*\text{NO}_2$ and $^*\text{NO}_3$ for Cu and V-Cu. (d) The relationship between d band center and electrode potential for Cu and V-Cu.

5. If the d-band model and the description of d-band center is still valid in electrified metals? Why?

Author response: We appreciate the valuable question from the reviewer. The d-band center model plays a crucial role in correlating the d-orbitals of transition metals and their adsorption behavior. Under varying electrode potentials, the adsorption behavior on the metal surface and the electronic structure of the metal itself will change, such as shifts in the Fermi level and the quantity of electrons transferring from the metal to the adsorbate.

Upon testing, we found that while the system is charged, the d-band center still effectively predicts adsorption strengths across different samples. However, its prediction accuracy wanes for a single sample under different electrode potentials.

It's understood that the cornerstone of the d-band center model suggests that active

sites with a more positive d-band center will exhibit stronger adsorption. Results from Fig. R3d illustrate that under electrode potentials ranging from -0.8 V to 0.4 V, the d-band center of V-Cu is consistently closer to the Fermi level compared to Cu. This successfully predicts the fact that V-Cu has a stronger adsorption energy than Cu (as seen in Fig. R3c). However, for the same sample under different electrode potentials, taking Cu adsorbing NO₃ as an example, as the electrode potential decreases, the d-band center becomes more negative, contradicting the predictions from the d-band center theory.

We deduced the following reasons to explain the effectiveness and limitations of the d-band center model:

a. **Predicting Adsorption Strengths Across Different Samples under the Same Electrode Potential:** When considering the adsorption strengths of different samples at the same electrode potential, the d-band center remains a powerful predictor. The d-band model is pivotal in elucidating the bonding and electronic behaviors between transition metals and adsorbates. Specifically, the interaction between the d-orbitals of transition metals and the p-orbitals of adsorbates results in the full occupation of bonding orbitals and partial occupation of antibonding orbitals. Interactions closer to the Fermi level between the d and p orbitals will lead to a decreased occupation of the antibonding orbitals and less weakening of the bonding orbitals, hence forming stronger chemical bonds. Under these conditions, such as when both Cu and V-Cu are at the same electrode potential, the basic molecular orbital interaction mechanism remains consistent, even when the system is charged. This makes the d-band center a valid descriptor under such circumstances.

b. **Predicting Adsorption Strength of a Single Sample under Different Electrode Potentials:** The predictive capacity of the d-band center might be challenged when trying to forecast the adsorption strength of a single sample under various electrode potentials. Artificially introducing or removing electrons from the system through the double-reference method to simulate different electrode potentials can lead to complex effects. For instance, as depicted in Fig. R4, altering the electron count will change the position of the Fermi level, which in turn affects the position of the d-band

center. This phenomenon is visually presented in Fig. R3d. Moreover, the d-band center model, based on the Newns-Anderson model, assumes the p-orbitals of the adsorbate remain unchanged. However, as the charge state of the system changes, the p-orbitals of the adsorbate might also be affected, complicating the d-p orbital interactions further.

c. **Conclusion:** Taking a broader perspective, even though the d-band center might face challenges under specific conditions like in charged systems, it remains one of the most successful and widely applied descriptors in the catalysis domain in recent years. Its pivotal role in predicting and elucidating numerous catalytic phenomena, especially its intuitive representation of the interactions between d-orbitals and adsorbate orbitals, is the foundation of its success. Typically, the d-band center is utilized to compare different samples and based on these comparisons, predict the adsorption energy of each sample for a specified molecule. In this context, the insights provided by the d-band center remain invaluable and robust.

Fig. R4. The effect of addition and removal of electrons on the d band center of the system.

Fig. R5. The relationship between the amount of charge doped and the Fermi level of the system for (a) Cu and (b) V-Cu.

6. The high selectivity of ammonia is the major achievements in this work. It was observed in the previous works it is quite difficult to achieve high ammonia selectivity at low overpotentials. However, the selectivity between nitrite and ammonia productions was not explained well. We need all kinetic barriers in fig. 4c and analyze the reaction rate by microkinetic modelling and make comparison between ammonia and nitrite productions.

Author response: We would like to sincerely thank the reviewer for offering this valuable insights, which is really significant for us to further strengthen our work. Your observations regarding the efficient selectivity towards ammonia and the challenges of achieving high selectivity at low overpotentials have certainly provided us with novel perspectives and avenues for consideration. Moreover, I concur wholeheartedly with your suggestion of delving deeper into the kinetic barriers presented in Fig. 4c and the imperative need to utilize microkinetic modeling to elucidate the reaction rates.

The constant potential simulations, potential-dependent activation energy calculations, CHE model, and the Slow-growth method for determining reaction activation energies (Method R1, R2, R3, and R5) are all encapsulated in Fig. 4. The defect site of V-Cu, characterized by its lower coordination number, is in a state of under-coordination, resulting in a notably more negative adsorption energy. Specifically, the adsorption free energies of V-Cu for nitrate and nitrite are -1.62 eV and -1.34 eV respectively, in stark contrast to the -0.33 eV and -0.714 eV of the Cu sample. This attests to V-Cu's superior capability in sequestering nitrate ions, concurrently inhibiting the desorption of NO₂ into the solution. The rate-determining step for both V-Cu and Cu occurs during the protonation of *NH₂OH, with activation energies of 0.93 eV and 1.07 eV respectively. V-Cu demonstrates an advantage in almost all kinetic barriers, emphasizing its enhanced NH₃ conversion capability. Using Method R4, we employed the MKMCXX software to execute microkinetic rate modeling for the NO₃RR of both V-Cu and Cu.

As illustrated in Supplementary Fig. 22, the reaction approaches equilibrium around t=1500s. Supplementary Fig. 22a elucidates the coverage analysis of key intermediates on Cu. Although Cu has a relatively weak nitrate adsorption energy, the

deoxygenation rate of $^*\text{NO}_3$ is sufficiently rapid to prevent its prolonged adherence to the site. The coverage of $^*\text{NO}_2$ remains zero throughout the reaction, implying that the scant $^*\text{NO}_2$ produced from $^*\text{NO}_3$ deoxygenation is either swiftly converted by the catalyst or directly desorbed into the solution. Upon reaching equilibrium, a fraction of $^*\text{-Cu}$ remains underutilized, while $^*\text{O}$ and $^*\text{OH}$ dominate nearly 60% of the sites. These observations stem from Cu's weaker adsorption energies coupled with its higher kinetic barriers.

Supplementary Fig. 22b portrays the coverage analysis of vital intermediates on V-Cu. Due to its strong nitrate adsorption, $^*\text{-V-Cu}$ is rapidly depleted. The negligible deoxygenation activation energy of nitrate and the lower protonation activation energies for $^*\text{O}$ and $^*\text{OH}$ on V-Cu lead to a rapid production and subsequent consumption of $^*\text{O}$. Subsequently, $^*\text{NO}_2$ almost entirely replaces $^*\text{-V-Cu}$, and by the time equilibrium is achieved, the coverage of $^*\text{NO}_2$ is nearly 100%. This underscores the possibility that the further conversion of $^*\text{NO}_2$ might be the linchpin affecting NH_3 production on V-Cu. It also indicates the heightened propensity of V-Cu to capture nitrite ions, ensuring they remain at the active sites for conversion rather than migrating into the solution.

In conclusion, as depicted in Supplementary Fig. 22c, the steady-state yields of nitrite and ammonia for Cu and V-Cu were determined. The results reveal that the ammonia yield for V-Cu stands at 1.186×10^{-7} mol/s, with the nitrite yield approximating 8.879×10^{-6} mol/s. Conversely, the ammonia yield for Cu is 1.289×10^{-9} mol/s, with nitrite production around 4.8×10^{-4} mol/s.

Supplementary Figure 22. Results of microkinetic modelling. The coverage of key intermediates during the reaction, namely * , $^*\text{NO}_3$, $^*\text{NO}_2$, $^*\text{O}$ and $^*\text{OH}$, as a function of time for (a) Cu and (b) V-Cu. (c) The reaction reaches the steady state, nitrite and ammonia productive rate for Cu and V-Cu.

Supplementary Figure 20. Free energy as a function of the CV of NO₃RR pathway by Slow-growth method for Cu.

Supplementary Figure 21. Free energy as a function of the CV of NO₃RR pathway by Slow-growth method for V-Cu.

7. The recent experiments (10.1002/anie.202303327) found the electrochemical performance of nitrate conversion can be enhanced by tandem process. If the present results are contributed from the second electroreduction of as-produced nitrite?

Author response: Thank you for bringing this important question to our attention regarding the possible secondary reduction of nitrite. Existing research indeed suggests that during the electrochemical reduction on copper-based materials, nitrate is first transformed into nitrite, which subsequently gets reduced to ammonia. Specifically, this

is documented in both Nat. Commun. 2022, 13, 7899 and Adv. Mater., 2023, 35, 2202952, where copper, as a component of the material, facilitates the conversion from nitrate to nitrite.

Our research corroborates this finding. By tracking the evolution of products over time, we have clearly observed the presence of nitrite during the reaction process (Fig. 2f). We speculate that, during the reduction of nitrate, some nitrite might be formed due to desorption. Simultaneously, we observed that V-Cu NAE exhibits a higher selectivity compared to defect-free Cu NWs. We hypothesize that the presence of vacancies might enhance the adsorption of nitrite, thereby efficiently promoting its reduction to ammonia. This observation aligns with our experimental data, where the concentration of nitrite gradually decreases to a very low level as the electrocatalytic reaction progresses.

Fig. 2. The electrocatalytic activity for NO₃RR in H-cell. (f) Time-dependent concentration change of NO₃⁻, NH₃, NO₂⁻ and FE over the V-Cu NAE at -0.3 V (vs. RHE).

8. The reviewer did not see any descriptions regarding the solvation effects of adsorbates. It seems all adsorption energies are simply calculated by models in vacuum?

Author response: We thank the reviewer for raising the question regarding solvation effects. Indeed, accounting for solvation effects is pivotal for accurate adsorption energy calculations and can substantially affect the overall results. Your question about whether we solely calculated adsorption energies in vacuum made me

realize that our original manuscript might not have adequately described this part of the methodology. We adopted a hybrid solvation model, which incorporates both implicit and explicit solvation effects, to account for the influence of solvent on the adsorption energies of intermediates. Specifically, the explicit solvation model introduces the local hydrogen bonding stabilization effects on the intermediates.

Supplementary Table 4. Parameters for fitting the relationship between E_{free} and U under constant potential simulation for Cu.

	a	b	c	C (eV)	U_0 (V/SHE)	E_0 (eV)
*NO ₃	-2.75	-0.68	-578.65	5.49	-0.12	-578.69
*NO ₂	-2.26	-1.07	-573	4.51	-0.24	-573.13
*NO	-2.41	-1.54	-566.75	4.81	-0.32	-567
*NOH	-2.64	-2.35	-570.62	5.28	-0.45	-571.14
*N	-2.21	-1.51	-560.41	4.42	-0.34	-560.67
*NHOH	-2.21	-1.81	-574.72	4.43	-0.41	-575.09
*NH ₂ OH	-2.65	-2.22	-578.55	5.29	-0.42	-579.01
*NH ₂ O	-2.44	-1.47	-574.37	4.87	-0.3	-574.59
*NHO	-2.31	-0.82	-570.45	4.62	-0.18	-570.53
*NH	-2.37	-2.14	-565.25	4.74	-0.45	-565.74
*NH ₂	-2.16	-1.77	-569.59	4.31	-0.41	-569.95
*NH ₃	-2.24	-2.61	-573.99	4.47	-0.58	-574.75
*H	-2.48	-2.07	-556.66	4.97	-0.42	-557.09
*O	-2.34	-1.81	-559.88	4.67	-0.39	-560.23
*OH	-2.52	-2.59	-564.19	5.04	-0.51	-564.86
slab	-2.47	-2.09	-552.53	4.93	-0.42	-552.97
*H ₂ O	-2.32	-2.68	-568.17	4.64	-0.58	-568.95
*H + *OH	-2.43	-2.04	-567.67	4.85	-0.42	-568.10

Supplementary Table 5. Parameters for fitting the relationship between E_{free} and U under constant potential simulation for V-Cu.

	a	b	c	C (eV)	U_0 (V/SHE)	E_0 (eV)
*NO ₃	-2.7	-1.14	-583.23	5.4	-0.21	-583.35
*NO ₂	-2.72	-1.31	-577.44	5.44	-0.24	-577.6
*NO	-2.63	-1.09	-571.48	5.26	-0.21	-571.6
*NOH	-2.44	-1.54	-557.38	4.89	-0.31	-557.62
*N	-2.74	-1.03	-564.98	5.48	-0.19	-565.08
*NHOH	-2.73	-2.42	-574.72	-579.24	-0.44	-575.26
*NH ₂ OH	-2.68	-2.36	-583.21	5.35	-0.44	-583.73
*NH ₂ O	-2.44	-1.47	-574.37	4.87	-0.3	-574.59
*NHO	-2.43	-1.35	-575.08	4.86	-0.28	-575.27
*NH	-2.22	-1.6	-569.77	4.45	-0.36	-570.06
*NH ₂	-2.54	-2.21	-573.84	5.08	-0.44	-574.33
*NH ₃	-2.78	-2.86	-578.23	5.55	-0.52	-578.97
*H	-2.53	-1.65	-561.30	5.06	-0.33	-561.57
*O	-2.34	-1.81	-559.88	4.67	-0.39	-560.23
*OH	-2.82	-2.55	-568.79	5.64	-0.45	-569.37
slab	-2.75	-1.92	-552.53	5.5	-0.35	-552.86
*H ₂ O	-2.83	-2.61	-572.64	5.64	-0.46	-573.24
*H + *OH	-3.44	-2.83	-572.62	6.88	-0.41	-573.21

Supplementary Table 6. Parameters used by Slow-growth approach for Cu.

Slow Growth Approach in Reaction Networks	CV (Å)	Transformation velocity (Å·fs ⁻¹)
*NO ₃ + * → *NO ₂ + *O	d(N-O)-d(Cu-O)	0.0003 (t=10000 fs)
*NO ₂ + * → *NO + *O	d(N-O)	0.0002 (t=10000 fs)
*NO + *H → *NOH + *	d(O-H)	-0.0002345 (t=10000 fs)
*NO + *H → *NHO + *	d(N-H)	-0.000185 (t=10000 fs)
*NOH + *H → *NHOH + *	d(N-H)	-0.000176 (t=10000 fs)
*NOH + *H → *N + H ₂ O + *	d(O-H)	-0.0002182 (t=10000 fs)
*NHO + *H → *NH ₂ O + *	d(N-H)	-0.0002018 (t=10000 fs)
*NHO + *H → *NHOH + *	d(O-H)	-0.0002676 (t=10000 fs)
*NHOH + *H → *NH ₂ OH + *	d(N-H)	-0.0001937 (t=10000 fs)
*NHOH + *H → *NH + H ₂ O + *	d(O-H)	-0.000172 (t=10000 fs)
*NH ₂ O + *H → *NH ₂ OH + *	d(O-H)	-0.000166 (t=10000 fs)
*NH ₂ OH + *H → *NH ₂ + H ₂ O + *	d(O-H)	-0.0002036 (t=10000 fs)
*N + *H → *NH + *	d(N-H)	-0.000188 (t=10000 fs)
*NH + *H → *NH ₂ + *	d(N-H)	-0.000218 (t=10000 fs)
*NH ₂ + *H → *NH ₃ + *	d(N-H)	-0.0002 (t=10000 fs)
*O + *H → *OH + *	d(O-H)	-0.00022 (t=10000 fs)
*OH + *H → *H ₂ O + *	d(O-H)	-0.0002856 (t=10000 fs)

Supplementary Table 7. Parameters used by Slow-growth approach for V-Cu.

Slow Growth Approach in Reaction Networks	CV (Å)	Transformation velocity (Å·fs ⁻¹)
*NO ₃ + * → *NO ₂ + *O	d(N-O)	0.0003 (t=4000 fs)
*NO ₂ + * → *NO + *O	d(N-O)	0.0003 (t=6000 fs)
*NO + *H → *NOH + *	d(O-H)	-0.000203 (t=10000 fs)
*NO + *H → *NHO + *	d(N-H)	-0.0001794 (t=10000 fs)
*NOH + *H → *NHOH + *	d(N-H)	-0.0002356 (t=10000 fs)
*NOH + *H → *N + H ₂ O + *	d(O-H)	-0.0002075 (t=10000 fs)
*NHO + *H → *NH ₂ O + *	d(N-H)	-0.0001202 (t=10000 fs)
*NHO + *H → *NHOH + *	d(O-H)	-0.0001619 (t=10000 fs)
*NHOH + *H → *NH ₂ OH + *	d(N-H)	-0.0001866 (t=10000 fs)
*NHOH + *H → *NH + H ₂ O + *	d(O-H)	-0.0002017 (t=10000 fs)
*NH ₂ O + *H → *NH ₂ OH + *	d(O-H)	-0.0001438 (t=10000 fs)
*NH ₂ OH + *H → *NH ₂ + H ₂ O + *	d(O-H)	-0.0002157 (t=10000 fs)
*N + *H → *NH + *	d(N-H)	-0.0001297 (t=10000 fs)
*NH + *H → *NH ₂ + *	d(N-H)	-0.0001886 (t=10000 fs)
*NH ₂ + *H → *NH ₃ + *	d(N-H)	-0.0001893 (t=10000 fs)
*O + *H → *OH + *	d(O-H)	-0.000175 (t=10000 fs)
*OH + *H → *H ₂ O + *	d(O-H)	-0.0001838 (t=10000 fs)

Reviewer #2: In the manuscript of “Defective Cu Enabled Triple Synergistic Modulation Endows Superior Electrochemical Ammonia Production at Wide-Range Nitrate Concentrations”, the defect-rich copper nanowires exhibited high current density and Faraday efficiency across a wide range of nitrate concentrations, reducing nitrate levels to meet WHO drinking water standards, and leading in performance. Through the authors’ detailed exploration on the triple synergistic modulation induced by copper defects, employing both synchrotron in-situ infrared and DFT calculations, authors have successfully unveiled the mechanisms that contribute to the superior electrochemical ammonia production. Finally, through a two-electrode system, authors achieved impressive nitrate degradation and ammonia production capabilities at low cell voltages, further expanding its application. Overall, the authors’ work provides inspiration for solving the degradation and ammonia production of nitrate pollutants under real conditions. I recommend the acceptance of this manuscript after minor revisions. The issues that need to be addressed are listed as follows:

Author response: Firstly, we would like to express our sincere gratitude to Reviewer for reviewer's thorough review and positive feedback on our manuscript titled, “Defective Cu Enabled Triple Synergistic Modulation Endows Superior Electrochemical Ammonia Production at Wide-Range Nitrate Concentrations”. We are heartened to know that Reviewer recognize the novelty and significance of our work, especially in relation to the degradation and ammonia production of nitrate pollutants under real conditions. Reviewer's acknowledgment of our detailed exploration, which combines both synchrotron in-situ infrared techniques and DFT calculations, further validates the depth of our research.

We appreciate Reviewer's recommendation for acceptance after minor revisions and assure reviewer that we will diligently address each point Reviewer listed in the subsequent comments. Our utmost goal is to enhance the quality and clarity of our manuscript, and we sincerely value Reviewer's expert insights to achieve this.

In response to reviewer's valuable feedback, we have implemented the following modifications:

1. **Cost Analysis:** We compared our electrocatalytic system's cost and environmental

benefits with the Haber-Bosch process.

2. Nitrate Concentration Performance: We evaluated V-Cu NAE against Cu NWs across varying nitrate concentrations and expanded this to other samples at different potentials.

3. Catalyst Durability: Through extended cycle tests and detailed pre/post characterizations, we assessed our catalyst's stability during nitrate reduction.

4. In-situ Data Enhancement: We improved clarity in Fig. 1j and 3a with distinct color labels, aiding result interpretation.

5. DFT Calculations & Diagrams: Our revised DFT calculations now incorporate an electrochemical perspective, and we've updated our free energy diagram for NO₃RR and HER.

6. Glycerol Oxidation Insights: We've detailed the yield and Faraday efficiency of glycerol oxidation at the anode.

The subsequent sections present a detailed response to reviewer's invaluable suggestions.

1. The field of electrocatalytic nitrate reduction to ammonia is very promising and active. In this research, have the authors considered the high energy consumption issue associated with the eight-electron transfer process typically required for nitrate ion reduction? Additionally, compared to the established and widely applied Haber-Bosch process, does the method have sufficient cost-effectiveness to offset this high electrical energy cost? If so, could authors provide supportive data or models to prove the cost-benefit advantages of the electrocatalytic nitrate reduction method? These are key factors in evaluating the feasibility of an electrocatalytic nitrate reduction method, and I look forward to authors' detailed response.

Author response: Thank reviewer for your insightful comments and feedback. The questions reviewer has raised are critical, focusing on the potential applicability and effectiveness of our research. We are delighted to respond to reviewer's queries and provide reviewer with more detailed explanations.

Reviewer first inquired about the high energy consumption associated with the

eight-electron transfer process in our research. Indeed, we have considered this issue from an angle as close to industrialization as possible. In our work, we adopted a two-electrode system to couple the actual wastewater nitrate reduction to ammonia and glycerol oxidation. To give reviewer a clear view of the energy consumption, the following calculations have been made:

Firstly, without voltage correction, we obtained a current of about 500 mA cm^{-2} at a voltage of -1.4 V , resulting in an ammonia production rate of about $30 \text{ mg h}^{-1} \text{ cm}^{-2}$. This data was obtained from our experiments using a two-electrode system.

Therefore, the power per square centimeter is $P = I V = 0.5 \text{ A} * 1.4 \text{ V} = 0.7 \text{ W}$, and the hourly output is 0.03 g . Hence, the electrical energy required to produce 1 g of ammonia can be calculated as:

$$(a) \text{ Energy per hour} = \text{Power} * \text{Time} = 0.7 \text{ W} * 1\text{hr} = 0.7 \text{ Wh}$$

$$(b) \text{ Energy to produce } 1\text{g ammonia} = \text{Energy per hour} / \text{Output per hour} = 0.7 \text{ Wh} / 0.03 \text{ g} = 23.3 \text{ Wh/g}$$

$$(c) \text{ Adjusted for efficiency (80\%)} = 23.3\text{Wh/g} / 0.80 = 29.1\text{Wh/g}$$

This can be extrapolated to produce 1 kg (i.e., 1000g) of ammonia:

$$(d) \text{ Energy to produce } 1 \text{ kg ammonia} = \text{Energy to produce } 1\text{g ammonia} * 1000\text{g} = 29.1\text{Wh/g} * 1000\text{g} = 29 \text{ kWh.}$$

Based on the price of renewable energy ($\text{US}\$0.03 \text{ kWh}^{-1}$), the electrical energy cost to produce 1kg of ammonia can be calculated:

$$(e) \text{ Cost to produce } 1\text{kg ammonia} = \text{Energy to produce } 1\text{kg ammonia} * \text{cost per kWh} = 29 \text{ kWh} * \text{US}\$0.03 \text{ kWh}^{-1} = 0.87 \text{ US\$}.$$

This is lower than the current price of ammonia ($1.0\text{-}1.5 \text{ USD per kg}$). However, it should be noted that this is a simple calculation of electrical energy costs from an industrial perspective, not considering equipment costs.

Considering the environmental benefits of removing nitrate pollutants and the valuable products produced at the anode, the two-electrode system we reported is very attractive.

Additionally, compared to the Haber-Bosch process, electrocatalytic nitrate reduction to ammonia has advantages such as simpler equipment, less environmental

pollution, and simpler reaction conditions. We have compared these factors between the two methods and included them in the following table.

Table R1: Comparison of Key Factors between Haber-Bosch Process and Electrocatalytic Nitrate Reduction (NO₃RR).

	Haber-bosch process	NO ₃ RR (Our work)
Ammonia Production Cost	US\$ 1.0 - 1.5	US\$ 0.8
Environmental Impact		Wastewater Treatment Capacity		Equipment Cost		Sustainability		Reaction conditions		
Note: In this table, a green smiling face indicates positive aspects, a yellow neutral face indicates neutral aspects, and a red crying face indicates negative aspects.

2. To demonstrate the performance enhancement of copper in a wide nitrate concentration range due to defect introduction, I hope authors could set up further comparisons, testing pure copper without defects across a broad range of nitrate concentrations. Moreover, the authors should also supplement the performance of other comparative samples at different potentials.

Author response: I appreciate reviewer's invaluable suggestions and comments. I fully agree with reviewer's perspective that we should further compare the performance of pure copper across a broad range of nitrate concentrations to underscore the role defects play in this wide range. We have also supplemented the performance of other comparative samples at different potentials as reviewer recommended. Reviewer is absolutely right that we need to present these comparative experiments in a more comprehensive and detailed way.

Accordingly, we have gathered the relevant data and added the results of these additional experiments in our revised manuscript. These new data not only further

confirm our hypothesis that the introduction of defects enhances the performance of copper, but also provide us with a deeper understanding of the performance of other samples at different potentials. In the main text, we have also carried out a detailed analysis and discussion of these new data. Please refer to the revised versions on page 8 and page 11. Note the highlighted sections at line 20 on page 8 and line 2 on page 11 where you can find the relevant modifications.

Supplementary Figure 8. Potential-dependent Faradaic efficiency and selectivity of ammonia over (a) Cu₃N NWs and (b) Cu NWs.

Supplementary Figure 9. Current density, FE of NH₃, selectivity of NH₃, and conversion of NO₃⁻ in Cu NWs under different concentration nitrate sources.

3. More experimental data to support catalyst durability would be appreciated. After continuous cycle tests, is the catalyst stable in the nitrate reduction process? Are there any changes in its phase and morphology? What are the reasons for catalyst performance degradation? More characterization and comparisons for the catalyst before and after measurements are needed.

Author response: Thank you for your insightful comments and suggestions. We appreciate reviewer's request for additional experimental data and clarification on the stability and durability of the catalyst involved in the nitrate reduction process. We acknowledge the importance of understanding any changes in the phase and morphology of the catalyst after repeated use.

In response to reviewer's suggestions, we have carried out additional cycle tests and performed detailed characterizations for the catalyst before and after the measurements. Please refer to the revised version on page 19, and note the highlighted section at line 9 where you can find the relevant modifications.

As shown in Supplementary Fig. 29, no changes in the phase of the catalyst were observed following the stability tests. It remained in the Cu phase.

Supplementary Figure 29. (a) XRD spectra of V-Cu NAE catalyst before and after 120 h electrocatalytic NO₃RR. (b) Enlarged XRD spectra of the boxed region from (a).

Subsequent to thorough examination via SEM, we observed several distinctive transformations following the cycling process. Specifically, certain nanowire arrays exhibited detachment from the copper foam scaffold, as demonstrated in Supplementary Fig. 30b and d - this provided a stark contrast to the pre-cycling state shown in Supplementary Fig. 30 a, c, e, and g. Moreover, we noticed a phenomenon of some nanowires aggregating together (Supplementary Fig. 30). Additionally, in certain instances, fractures within individual nanowires were also evident (Supplementary Fig. 30). These observed changes are likely the contributing factors to the observed performance degradation during the cycling process.

Supplementary Figure 30. SEM images of the V-Cu NAE catalyst before (a, c, e, g) and after 120 h (a, c, e, g) electrocatalytic NO₃RR.

4. The authors have performed some in-situ characterizations, such as synchrotron in-

situ infrared, in situ Raman spectroscopy, which are very valuable for research as they can intuitively display the inherent characteristics of substances under specific conditions. However, presenting these data in the form of color mapping (Fig. 1 j and Fig. 3 a), without color labels for specific numerical expression, may lead to loss or misinterpretation of information.

Author response: I greatly appreciate reviewer's insightful comment regarding the presentation of our in-situ characterization data, namely synchrotron in-situ infrared and in-situ Raman spectroscopy results. reviewer's remark about the potential loss or misinterpretation of information due to the lack of color labels for specific numerical expression in our color mapping (Fig. 1j and Fig. 3a) is indeed correct and we are grateful for reviewer's suggestion.

We understand the importance of presenting our data as clearly and unambiguously as possible. Consequently, we have updated Fig. 1j and 3a in our revised manuscript to include color labels that correspond to specific numerical values. We believe these modifications will facilitate a better understanding and interpretation of our data.

Fig. 1. (j) In-situ Raman spectroscopy of the electrochemical reduction of Cu₃N to V-Cu and its contour map corresponding to the 600-700 cm⁻¹ range.

Fig. 3. Operando Synchrotron radiation-FTIR spectroscopy measurements under various potentials for V-Cu NAE during NO₃RR. (a) Three-dimensional FTIR spectra and corresponding contour maps in the range of 4000-1250 cm⁻¹.

5. From a reviewer's perspective, it is crucial to reflect the use of an electrochemical model in DFT calculations, rather than relying solely on thermodynamics calculations. Specifically, it's important to account for the significant influence of the electrode potential on the rate-determining energy barrier for NO₃⁻ reduction reaction (NO₃RR). This reaction involves the adsorption of NO₃⁻ and proton - electron transfer, both of which exhibit contrasting responses to potential. With this in mind, the authors should indicate the voltage corresponding to the experimentally optimal performance, and present a free energy diagram of NO₃RR from an electrocatalytic viewpoint. This would greatly aid readers in gaining a comprehensive understanding of the process, and in evaluating the effectiveness and applicability of the model. As a reviewer, I perceive this to be a critical component of understanding the research results, and also a reflection of the depth and quality of the author's investigation. Furthermore, the energies calculated for all intermediates should be included in the supplementary information tables.

Author response: First and foremost, we sincerely appreciate the valuable comments you've provided for our manuscript.

Regarding your point on the significance of the electrochemical model in DFT calculations, we are in complete agreement. Relying solely on thermodynamic calculations doesn't suffice to capture the intricacies of the NO_3^- reduction reaction process. To address this, we have now calculated the free energy changes of NO_3RR and HER at an electrode potential of -0.3 V. Furthermore, we've revised the Reaction pathways and free energy diagram to detail the adsorption configurations of various intermediates generated during HER and NO_3RR over V-Cu NAE and Cu NWs. Please refer to the revised version on page 14, and note the highlighted section at line 1 where you can find the relevant modifications.

For the convenience of readers and to provide a comprehensive understanding, all energies calculated for intermediates, as well as the energies for transition states, are now included in the supplementary information tables.

Once again, thank you for your insightful suggestions. We highly value them and look forward to any further feedback on our revised manuscript.

Supplementary Table 8. Calculated Gibbs free energies of NO_3RR at -0.289 V vs RHE and $T = 298.15$ K for Cu and V-Cu.

System	V-Cu	Cu
	Free energy (eV)	Free energy (eV)
* NO_3	-1.62	-0.33
* NO_2	-3.09	-2.46
* NO	-4.70	-4.82
* NOH	-4.50	-4.38
* N	-6.26	-6.59
* NHOH	-5.37	-4.48
* NH_2OH	-5.62	-4.83
* NH_2O	-5.46	-4.81
* NHO	-5.70	-4.56
* NH	-7.17	-6.76
* NH_2	-8.20	-7.08

*NH ₃	-8.35	-7.59
NO ₂ ⁻	-1.75	-1.75
NH ₃	-7.78	-7.78
*H ₂ O	-0.517	-0.286
*H+*OH	-0.824	-0.734
*H	-0.840	-0.636

Supplementary Table 9. Activation energies of NO₃RR at -0.289 V vs RHE and T = 298.15 K for Cu and V-Cu.

Reaction	V-Cu	Cu
	E _a (eV)	E _a (eV)
*NO ₃ + * → *NO ₂ + *O	0.010	0.689
*NO ₂ + * → *NO + *O	0.925	0.727
*NO + H → *NHO	0.618	0.958
*NO + H → *NOH	1.344	1.069
*NOH + H → *NHOH	1.093	0.887
*NOH + H → *N + H ₂ O	1.620	1.168
*NHO + H → *NHOH	0.745	0.797
*NHO + H → *NH ₂ O	0.434	0.954
*NHOH + H → *NH + H ₂ O	1.198	0.842
*NHOH + H → *NH ₂ OH	1.051	0.828
*NH ₂ O + H → *NH ₂ OH	1.000	0.661
*NH ₂ OH + H → *NH + H ₂ O	1.349	1.069
*N + H → *NH	0.371	0.633
*NH + H → *NH ₂	1.072	0.815
*NH ₂ + H → *NH ₃	0.920	0.755
*O + *H → *OH + *	1.232	0.892
*OH + H → H ₂ O + *	1.199	0.936
*H ₂ O + * → *H+*OH	-0.135	0.434

6. The authors integrated the nitrate reduction reaction and glycerol oxidation reaction in wastewater through a two-electrode system and provided detailed analysis of nitrate reduction at the cathode end. However, analysis on the yield and Faraday efficiency of glycerol oxidation at the anode end is missing. Furthermore, during the gas stripping process, the authors used air to blow out the ammonia product instead of using an inert gas. Could this have an effect on the collection of the ammonia product?

Author response: Thank reviewer for reviewer's insightful comments and suggestions. We appreciate reviewer's keen observation regarding the lack of a detailed analysis of the yield and Faraday efficiency of glycerol oxidation at the anode end. In response to this, we have supplemented the necessary analysis and related data.

Supplementary Figure 28. A standard curve was generated by plotting the integral area of HCOO^- - ^1H relative to a specific peak of DMSO against the ^1H concentration in formate ion.

We conducted quantitative Nuclear Magnetic Resonance (NMR) measurements (Supplementary Fig. 28), which revealed that the Faraday efficiency of the glycerol oxidation reaction in our two-electrode system is 81.3%, and the yield is $0.01494 \text{ mol cm}^{-2} \text{ h}^{-1}$. We will include these findings in our revised manuscript. Please refer to the revised version on page 19, and note the highlighted section at line 4 where you can

find the relevant modifications.

Regarding reviewer's point about using air instead of an inert gas during the gas stripping process, reviewer have raised a valid concern. We chose to use air to strip out the ammonia product because it is more cost-effective and closely mimics the actual industrial conditions of the ammonia purification process, which typically utilizes air. While it's true that using air could potentially result in the formation of ammonium carbonate, it's worth noting that ammonium carbonate is unstable at high temperatures and decomposes back into ammonia.

Reviewer #3: In this study, a defect-rich Cu nanowire array electrode (V-Cu NAE) was fabricated, which exhibited superior performance in the electrochemical nitrate reduction reaction (NO₃RR) across a wide range of NO₃⁻ concentrations (1-100 mM) with high FEs, NH₃ yields, and NO₃⁻ conversions. In-situ characterization and DFT calculations indicate that the triple synergistic modulation on the defective Cu sites simultaneously enhances nitrate adsorption, promotes water dissociation, and suppresses hydrogen evolution, resulting in this high performance. Additionally, a two-electrode system was established, which integrated the NO₃RR in industrial wastewater with the glycerol oxidation reaction, further demonstrating its practicality. The manuscript is well organized, and the results are credible. However, some data still deserves further analysis. To meet the high quality of Nat. Commun., major revisions are required. Here are some comments for authors:

Author response: Firstly, we would like to express our profound gratitude for reviewer's thorough review of our manuscript and for the in-depth feedback reviewer provided. The insights and detailed suggestions reviewer highlighted are vital aspects that need further emphasis and refinement in our research. We highly appreciate the effort the reviewer has put forth to ensure our work aligns with the high-quality standards of Nat. Commun..

In response to reviewer's valuable feedback, we have implemented the following modifications:

1. Regarding the interpretation of the operando SR-FTIR spectra: We've verified that our measurements were conducted after background subtraction and have provided additional curves at open-circuit potential to illustrate this.
2. On the clarity of the images: To represent defects more clearly, we selected different regions for analysis. Additionally, we have optimized the type and color of the labels.
3. Concerning the absence of Cu NWs XPS data: Based on reviewer's suggestion, we performed supplementary Cu NWs XPS tests to further elucidate the influence of Cu vacancies on the sample's surface properties.
4. About the specificity of the electrochemical test conditions: We have refined the electrochemical testing conditions in the revised manuscript.

5. Pertaining to the repeated references to WHO drinking water standards: We have removed the repetitive mentions in the revised draft.

6. Regarding the additional characterization of samples after stability testing: We have conducted extra cyclic tests and detailed characterizations to understand any phase and morphological changes in the catalyst after repeated use.

We once again appreciate reviewer's invaluable feedback and time. We hope that our revised manuscript now meets the expectations of Nat. Commun. and look forward to reviewer's further feedback.

The subsequent sections present a detailed response to reviewer's invaluable suggestions.

1. The interpretation of the operando SR-FTIR spectroscopy (Fig. 3) could benefit from additional evidence to support it. It is particularly worth discussing that the typical absorption peak of water is located around 3400 cm^{-1} . Therefore, it is important to consider the possibility of water absorptions in this region and distinguish them from the free ammonia signals. Conducting the same test in a solution without NO_3^- reactants can indeed help to distinguish the contribution of NH_3 in the observed spectra.

Author response: We appreciate the reviewer's astute observations and constructive feedback on the interpretation of the operando SR-FTIR spectroscopy. The suggestion to consider the water absorptions around 3400 cm^{-1} was particularly pertinent. Recognizing the potential overlap between this peak and the free ammonia signals is a significant point that can impact the accuracy of our interpretations.

We apologize for the oversight in not explicitly mentioning in the manuscript that the in-situ infrared analysis (SR-FTIR) was conducted after background subtraction, which may have led to confusion for the reviewer. To address this, we have made sure to explicitly mention in our revised manuscript that the test was conducted post-background subtraction. Please refer to the revised version on page 12, and note the highlighted section at line 8 where you can find the relevant modifications.

To further elucidate our experimental method and address the queries raised by the reviewer, we have incorporated a new figure (Supplementary Fig. 14) in the

supplementary material of the revised manuscript. This figure clearly delineates our testing procedure, showcasing the background spectrum, the open-circuit potential (OCP) spectrum post-background subtraction, and the electrolysis spectrum at -0.3 V. Through this side-by-side representation of the spectra, we aim to provide readers with a lucid understanding of the background subtraction process. After accounting for the background, virtually no infrared peaks from water or other potential interferences are evident, ensuring a more accurate and intuitive depiction of the produced species. Additionally, we have integrated the OCP spectrum post-background subtraction in Fig. 3, enabling a more illustrative portrayal of species variation with the change in potential. Please refer to the revised version on page 13, and note the highlighted section at line 2 where you can find the relevant modifications.

We appreciate the reviewer's feedback and have made efforts to address the concerns raised. We hope our revisions provide clarity and resolve any outstanding issues.

Supplementary Figure 14. Infrared spectra illustrating the background, post-background subtraction at open-circuit potential (OCP), and, where indicated, post-background subtraction at -0.3 V. (a) Range: 1700-1000 cm⁻¹. (b) Range: 2900-3500 cm⁻¹. (c) Range: 4000-1250 cm⁻¹ (Excludes -0.3 V post-subtraction)

Fig. 3. Operando Synchrotron radiation-FTIR spectroscopy measurements under various potentials for V-Cu NAE during NO₃RR. (a) Three-dimensional FTIR spectra and corresponding contour maps in the range of 4000-1250 cm⁻¹. (b) Infrared signals in the range of 3000-4000 cm⁻¹. (c) Infrared signals in the range of 1700-1250 cm⁻¹.

2. Although images may undergo compression when converted into documents, some images remain excessively blurry. For example, the yellow and red arrows in Fig. 1e are not prominent. Moreover, the small white circle mentioned on page 7, line 105, may lead to misunderstanding. It is suggested to replace it with a different color to enhance contrast.

Author response: Thank you for your thoughtful feedback regarding the clarity of the defects in the spherical aberration electron microscopy images. In our effort to provide clearer visualization, we've captured new spherical aberration electron microscopy images. Additionally, we've color-coded the atoms for enhanced clarity. The white circles now represent defects. In response to your feedback, we have chosen distinct regions for analysis to depict the defects more explicitly. We've also modified

the colors and types of our markers to make them more distinguishable, as illustrated in Fig. 1e. Based on this updated visualization, we performed an intensity contour analysis on the atoms situated between the arrows in regions i, ii, and iii, showcased in Fig. 1f. Please refer to the revised version on page 7, and note the highlighted section at line 14 where you can find the relevant modifications. We sincerely appreciate your suggestions, which have significantly contributed to the improvement of our work.

Fig. 1. (e) Double Spherical Aberration-corrected HAADF-STEM image of V-Cu NAE, with the white circled areas indicating Cu vacancies. (f) Intensity profile of the Cu atoms between the two arrows in regions i, ii and iii of (e).

3. To analyze the impact of Cu vacancies on the surface properties of the samples, it is necessary to supplement the XPS data of Cu NWs that without vacancies.

Author response: We thank the reviewer for the invaluable feedback and recommendations. Heeding reviewer's suggestion, we conducted XPS tests on Cu NWs (copper nanowires without vacancies) and V-Cu NAE (copper nanowires with vacancies). The data clearly highlighted the transformative influence of Cu vacancies on the local electronic environment. Notably, the Cu $2p_{3/2}$ and Cu $2p_{1/2}$ peaks of V-Cu NAE shifted by 0.13 eV and 0.17 eV, respectively, relative to Cu NWs. Such changes underscore a pronounced interaction between the metal vacancies and electrons. Please refer to the revised version on page 8, and note the highlighted section at line 6 where you can find the relevant modifications.

Supplementary Figure 6. (a) XPS spectra of Cu NWs, V-Cu NAE, along with their magnified views in Cu 2p_{3/2} (b) and Cu 2p_{1/2} (c).

4. The electrochemical testing conditions, such as the applied potential, are not clearly stated after Fig. 2b. Detailed information needs to be provided to enhance the rigor of the study.

Author response: We recognize and apologize for the oversight in omitting some of the electrochemical testing conditions after Fig. 2b. In the revised manuscript, we will provide a comprehensive description of these conditions, including the applied potential and other essential parameters such as reaction time and stirring speed. For the reviewer's convenience, these additions have been highlighted in yellow within the revised text. Reviewer's constructive feedback is invaluable in ensuring the clarity and rigor of our study.

5. The WHO regulations for drinking water have been mentioned repeatedly.

Author response: Thank reviewer for pointing out the repeated mentions of the "WHO

regulations for drinking water." We acknowledge this oversight and in the revised manuscript, we have removed the repeated mentions in the sections concerning the two electrodes. We hope this revision addresses reviewer's concern.

6. The results of SEM, XRD or other related characterizations of the samples after stability test should also be provided. Evidence that can reflect the changes in Cu vacancies should be carefully analyzed.

Author response: We thank you for your insightful comments and suggestions. We recognize the importance of reviewer's request for SEM, XRD, and other related characterizations of the samples after the stability test. Understanding the phase and morphology changes of the catalyst, as well as the changes in Cu vacancies, after repeated use in the nitrate reduction process is crucial.

In response to reviewer's suggestions, we conducted additional cycle tests and performed detailed SEM, XRD, and XPS analyses of the catalyst before and after the measurements. Following reviewer's valuable advice, we used the XPS data of Cu NWs without vacancies as a reference to carefully analyze the changes in Cu vacancies before and after the cycle tests. Please refer to the revised version on page 19, and note the highlighted section at line 9 where you can find the relevant modifications.

As illustrated in Supplementary Fig. 29, no evident changes in the phase of the catalyst were observed following the stability tests. It remained in the metallic Cu phase.

Supplementary Figure 29. (a) XRD spectra of V-Cu NAE catalyst before and after 120 h electrocatalytic NO₃RR. (b) Enlarged XRD spectra of the boxed region from (a).

Subsequent to thorough examination via SEM, we observed several distinctive transformations following the cycling process. Specifically, certain nanowire arrays exhibited detachment from the copper foam scaffold, as demonstrated in Supplementary Fig. 30 b and d - this provided a stark contrast to the pre-cycling state shown in Supplementary Fig. 30 a, c, e, and g. Moreover, we noticed a phenomenon of some nanowires aggregating together (Supplementary Fig. 30f). Additionally, in certain instances, fractures within individual nanowires were also evident (Supplementary Fig. 30h). These observed changes are likely the contributing factors to the observed performance degradation during the cycling process.

Supplementary Figure 30. SEM images of the V-Cu NAE catalyst before (a, c, e, g) and after 120 h (b, d, f, h) electrocatalytic NO₃⁻RR.

Building on your earlier suggestion that XPS can reflect the impact of Cu vacancies on the surface properties of the samples, we utilized this approach to carefully analyze the changes in Cu vacancies throughout the cycling process, as reviewer

recommended. To this end, we conducted XPS tests on the following samples: Cu NWs (copper nanowires without vacancies), V-Cu NAE (copper nanowires with vacancies), and V-Cu NAE-cycle 120 h (post-120 h cycling samples of the V-Cu NAE). XPS analysis indicated that, relative to the pristine V-Cu NAE, the Cu peaks of the cycled V-Cu NAE still displayed a significant shift when juxtaposed with those of the defect-free Cu NWs. This persistent shift corroborates the enduring stability of the Cu vacancies throughout the cycling process, underscoring the sustained presence of Cu vacancies in the catalyst. We believe this finding effectively demonstrates the changes in Cu vacancies and hope that this clarification addresses reviewer's concern.

Supplementary Figure 31. (a) XPS spectra of Cu NWs, V-Cu NAE and V-Cu NAE after 120 hours of cycling, along with their magnified views in Cu 2p_{3/2} (b) and Cu 2p_{1/2} (c).

REVIEWER COMMENTS

Reviewer #1 (Remarks to the Author):

The authors have performed lots of calculations to improve understanding of the present experiments by high-quality theoretical methods. I recommend its publication.

Reviewer #2 (Remarks to the Author):

The authors have well addressed the points raised by reviewers and the revised paper is ready for acceptance.

Reviewer #3 (Remarks to the Author):

In the revised manuscript, the authors have properly addressed most of the questions raised by the reviewers. However, before publication in Nat. Commun., there remain some details to be resolved.

1. The issue regarding SR-FTIR spectroscopy has not been adequately addressed. Although the authors emphasized the standardization of their testing process, this deviates from our concerns. Despite the background subtraction at OCP, water still has the potential to accumulate on the catalyst surface, resulting in a peak around 3400 cm^{-1} when a specific voltage is applied (such as -0.3 V as demonstrated by the authors). Hence, we strongly recommend conducting a control test devoid of NO_3^- to ascertain the attribution of the infrared peak.

2. There are a few logical issues in the details of the manuscript. For example, at page 16 line 264, the authors said that "The low-coordinated V-Cu(111) showcases enhanced adsorption capabilities for nitrate and nitrite, with adsorption free energies of -1.62 eV and -1.34 eV , respectively. In comparison to Cu (-0.33 eV and -0.71 eV), V-Cu(111) captures nitrate from the solution more swiftly, while also suppressing nitrite adsorption". This appears to be contradictory.

3. In the file named Responds to Reviewers, the authors said that the adsorption of nitrate will be suppressed at more negative reduction potentials at page 9. However, at page 19, the authors also mentioned that the adsorption strength of both Cu and V-Cu towards NO_3 and NO_2 progressively intensifies as the electrode potential decreases. So how should we understand this difference?

Responds to Reviewers

Reviewer #1: The authors have performed lots of calculations to improve understanding of the present experiments by high-quality theoretical methods. I recommend its publication.

Author response: Thank you very much for your professional opinions and suggestions, particularly regarding the theoretical calculations in our paper. Your detailed analysis has significantly enhanced the precision and depth of this section, for which we are deeply grateful.

Reviewer #2: The authors have well addressed the points raised by reviewers and the revised paper is ready for acceptance.

Author response: Thank you for acknowledging our response to the review comments and for considering our revised paper ready for publication. Your feedback has greatly contributed to the improvement of our work, for which we are deeply grateful. Your support and guidance are immensely valuable to us.

Reviewer #3: In the revised manuscript, the authors have properly addressed most of the questions raised by the reviewers. However, before publication in Nat. Commun., there remain some details to be resolved.

Author response: First and foremost, I would like to express my sincere gratitude for the continued attention and recognition you have given to our research. The issues and suggestions you raised in your previous review have been invaluable to our study, guiding us towards deeper reflection and necessary improvements.

With your constructive feedback in mind, we now address the specific concerns raised in your latest comments as follows:

1. The issue regarding SR-FTIR spectroscopy has not been adequately addressed. Although the authors emphasized the standardization of their testing process, this deviates from our concerns. Despite the background subtraction at OCP, water still

has the potential to accumulate on the catalyst surface, resulting in a peak around 3400 cm^{-1} when a specific voltage is applied (such as -0.3 V as demonstrated by the authors). Hence, we strongly recommend conducting a control test devoid of NO_3^- to ascertain the attribution of the infrared peak.

Author response: Thank you for your valuable suggestions. Following your guidance, we conducted a control experiment without nitrate (Supplementary Fig. 14). The results indeed confirmed that at a specific voltage, the accumulation of water on the catalyst surface forms a distinct peak at 3400 cm^{-1} .

We have classified the peak at 3400 cm^{-1} as a water peak and, following your suggestions, made corresponding modifications to our manuscript, including the addition of the control experiment. Please refer to the revised version on page 12, and note the highlighted section at line 7, as well as on page 13 at line 1, where you can find the relevant modifications. This ensures the accuracy of our experimental results and the integrity of our research. We believe these changes will enhance the quality of our study.

Fig. 3. Operando Synchrotron radiation-FTIR spectroscopy measurements under various potentials for V-Cu NAE during NO_3RR .

Supplementary Fig. 14. Operando Synchrotron Radiation-FTIR spectroscopy measurements under various potentials for V-Cu NAE during Electrolysis without NO_3^- .

2. There are a few logical issues in the details of the manuscript. For example, at page 16 line 264, the authors said that “The low-coordinated V-Cu(111) showcases enhanced adsorption capabilities for nitrate and nitrite, with adsorption free energies of -1.62 eV and -1.34 eV , respectively. In comparison to Cu (-0.33 eV and -0.71 eV), V-Cu(111) captures nitrate from the solution more swiftly, while also suppressing nitrite adsorption” . This appears to be contradictory.

Author response: We sincerely appreciate your careful attention in pointing out the error in our manuscript on page 16, line 264. In the original text, we incorrectly stated “V-Cu(111) also suppresses nitrite adsorption,” when it should indeed read “V-Cu(111) also suppresses nitrite desorption.” This was an inadvertent typographical error, for which we deeply apologize and have now corrected in the manuscript. Please refer to the revised version on page 16, and note the highlighted section at line 10 where you can find the relevant modifications.

We have made the necessary revisions to the manuscript to ensure its accuracy and clarity. We appreciate your thorough review and valuable suggestions.

3. In the file named Responds to Reviewers, the authors said that the adsorption of nitrate will be suppressed at more negative reduction potentials at page 9. However, at page 19, the authors also mentioned that the adsorption strength of both Cu and V-Cu towards NO_3^- and NO_2^- progressively intensifies as the electrode potential decreases. So how should we understand this difference?

Author response: Thank you for your review and the inquiries raised regarding our paper. In response to your questions about the impact of electrode potential on the adsorption behavior of nitrates and nitrites, we provide a more detailed explanation here to clearly elucidate this complex scientific issue.

Under the constant potential model, the adsorption free energy of nitrate and

nitrite can be obtained by the following formula:

$$\Delta G_{free} = \Delta E_{free}(U) + \Delta ZPE - \Delta TS - eU \quad (S1)$$

Here, ΔE_{free} is the free energy under the grand canonical ensemble, ZPE is the zero-point energy, T is the temperature, S is the entropy, and U is the applied voltage with respect to the RHE.

Obviously, the effect of the applied voltage on ΔG_{free} is two parts, one part comes from the free energy under the potential dependent grand canonical ensemble, and the other part comes from $-eU$ under the CHE model. We merely state on Pages 9 and 19 what are the trends in the contribution of each to ΔG_{free} .

In page 9, we stated that the adsorption energy of nitrate would become more correct at a more negative reduction potential, referring here only to the contribution of $-eU$. When U is negative, $-eU$ will be positive, so that ΔG_{free} will become more positive.

In page 19, we state that the adsorption energies of nitrate and nitrite will become more negative at more negative reduction potentials. Under the constant charge model, ΔE_{DFT} is usually used instead of ΔE_{free} , while ΔE_{DFT} is potential independent, which is unreasonable. The constant potential model points out that $\Delta E_{free}(U)$ is still affected by the applied potential and has a quadratic relationship with U:

$$E_{free}(U_{SHE}) = -\frac{1}{2}C(U_{SHE} - U_{PZC})^2 + E_{PZC} = aU_{SHE}^2 + bU_{SHE} + c \quad (S2)$$

Near page 19, we mention the following formula to calculate the adsorption energies of nitrate and nitrite:

$$E_{ads}(*NO_x, x = 2,3) = E_{free}(*NO_x, U_{SHE}) - E_{free}(slab, U_{SHE}) - E_{DFT}(NO_x(g)) \quad (S3)$$

Where, $E_{free}(*NO_x, U_{SHE})$ and $E_{free}(slab, U_{SHE})$ are quadratic functions with U_{SHE} , and $E_{DFT}(NO_x(g))$ is a constant. So $E_{ads}(*NO_x, x = 2,3)$ still exhibits quadratic function with U_{SHE} . By substituting C, U_{PZC} , E_{PZC} (see Supplementary Table 4 and Supplementary Table 5) at the fitting into Formula S3, it is obtained that $E_{ads}(*NO_x, x = 2,3)$ is still a quadratic function with negative quadratic coefficients.

So at more negative voltages, $E_{ads}(*NO_x, x = 2,3)$ is more negative.

Synthesizing the contributions of the voltages mentioned in page 9 and page 19 to the ΔG_{free} , the results of Fig. 4d and Fig. 4e are obtained.

Fig. 4. DFT Calculations. (d) The NO₃RR pathways for Cu are displayed, with the optimal pathway distinctly highlighted in blue at $U = -0.289$ V vs RHE. (e) The NO₃RR pathways for V-Cu are demonstrated with the optimal pathway prominently marked in red at $U = -0.289$ V vs RHE.

Once again, we appreciate the time and effort you have dedicated to reviewing our work. We hope that the detailed explanations provided have adequately addressed your questions. Your feedback is highly valued, and we look forward to receiving any further comments or suggestions you may have.

REVIEWERS' COMMENTS

Reviewer #3 (Remarks to the Author):

In the revised manuscript, the authors have properly addressed all the questions raised by the reviewers. As a result, this paper is recommended to publication in Nat. Commun. with no need to further review.